# Continuous tuning of persistent luminescence wavelength by intermediate-phase engineering in inorganic crystals

Xin Zhang [1,4], Hao Suo[1,2,4], Yang Guo[1], Jiangkun Chen[1], Yu Wang[2], Xiaohe Wei[1], Weilin Zheng[1], Shuohan Li[1] & Feng Wang [1,3] ✉

Multicolor tuning of persistent luminescence has been extensively studied by deliberately integrating various luminescent units, known as activators or chromophores, into certain host compounds. However, it remains a formidable challenge to fine-tune the persistent luminescence spectra either in organic materials, such as small molecules, polymers, metal-organic complexes and carbon dots, or in doped inorganic crystals. Herein, we present a strategy to delicately control the persistent luminescence wavelength by engineering sub-bandgap donor-acceptor states in a series of single-phase Ca(Sr)ZnOS crystals. The persistent luminescence emission peak can be quasi-linearly tuned across a broad wavelength range (500–630 nm) as a function of Sr/Ca ratio, achieving a precision down to ~5 nm. Theoretical calculations reveal that the persistent luminescence wavelength fine-tuning stems from constantly lowered donor levels accompanying the modified band structure by Sr alloying. Besides, our experimental results show that these crystals exhibit a high initial luminance of 5.36 cd m$^{-2}$ at 5 sec after charging and a maximum persistent luminescence duration of 6 h. The superior, color-tunable persistent luminescence enables a rapid, programable patterning technique for high-throughput optical encryption.

Color tuning is a constant research objective in luminescent materials[1–6]. Multicolor emitters with tunable photophysical properties can fulfill various practical applications ranging from lighting[7–9] and display[10] to information security[11,12] and medical diagnosis[13,14]. Persistent luminescence (PersL) is an intriguing phenomenon in which materials continuously emit light after the cessation of excitation[15]. Because of the merits of long-lived, self-sustained, and interference-free light emission, PersL materials have emerged as a powerful tool for technological applications in various areas. Specifically, these materials show great potential as a biomarker for medical imaging[16,17], a signal indicator for emergency display[18], as well as a storage medium for electromagnetic radiation[19,20], thermal field[21], and mechanical action[22,23].

PersL has been realized in various material systems encompassing organic compounds and inorganic crystals. For example, the promotion and stabilization of triplet excitons via heavy metal doping[24] and/or rigid structure construction (e.g., crystallization, encapsulation into a matrix, host-guest doping, H-aggregation, etc.) endow organic phosphors with room-temperature PersL (also known as ultra-long phosphorescence)[25–27]. By further modulating the molecule compositions and conformations, multicolor PersL is achieved in a single host of organic materials, including crystalized single-component molecules[1], host-guest complexes[28], and polymers[29]. As a promising alternative, inorganic PersL materials have gained more commercial success than their organic counterparts, ascribed to their

[1]Department of Materials Science and Engineering, City University of Hong Kong, Hong Kong SAR, China. [2]College of Physics Science & Technology, Hebei University, Baoding 071002, China. [3]Hong Kong Institute for Clean Energy, City University of Hong Kong, Hong Kong SAR, China. [4]These authors contributed equally: Xin Zhang, Hao Suo. ✉e-mail: fwang24@cityu.edu.hk

overwhelming chemical/physical stability and PersL efficiency on top of highly tunable emission by rare-earth/transition-metal doping[30].

Until now, light emission in PersL materials has still been dominated by guest chromophores/luminescent ions occurring at discrete wavelengths[31,32]. Although color tuning of PersL in these systems can be implemented by controlling the composition/combination of the emission centers[32,33], the strategies inevitably leave spectral gaps in specific wavelength ranges, hindering the full-spectrum expression of PersL (Fig. 1a and Supplementary Table 1). On a side note, PersL emitters of various emission colors usually exhibit unbalanced brightness and efficiency[34], impeding their applications for multicolor displays and multiplexed bioimaging[29]. Therefore, it is highly desired to develop wavelength-tunable PersL of comparable emission intensity, preferably within a single material system, to avoid compatibility issues when incorporating various PersL materials into a single device.

Herein, we present a strategy for fine-tuning PersL emission by blending two isostructural compounds. We show that the structural consistency between CaZnOS and SrZnOS permits the construction of $Ca_{1-x}Sr_xZnOS$ ($x = 0-1$) solid solutions with linearly evolving physical properties for sub-bandgap donor state engineering. Accordingly, we demonstrate highly controllable and linearly shifted donor-acceptor (D-A) pair PersL by simply varying the $Sr^{2+}/Ca^{2+}$ ratio in the host materials (Fig. 1b). Specifically, we obtain a chain of PersL emitters with emission peaks successively varied from 500 to 630 nm in a precise step down to 5 nm. Based on the unprecedented PersL materials, a rapid, programmable patterning technique is established for high-throughput optical encryption.

## Results

We use a doping strategy to generate hybrid materials as it is an effective technique for material property modification[35,36]. The study by Tu et al. unveiled D-A emission characteristics through their research on the mechanical quenching of PersL in CaZnOS:Cu[37,38]. Concurrently, our prior investigations have demonstrated that in CaZnOS, copper and rare-earth elements (e.g., Y, Gd, Tb, Nd, Er, Ho, Tm, Dy, and Pr) are capable of forming D-A pairs. This interaction facilitates band emissions, which occur alongside the intrinsic lanthanide luminescence[21]. Consequently, in the current study, we employed Cu and non-luminescent Y as dopants to exclusively generate D-A emission for PersL modulation. The sub-bandgap D-A states established by $Cu^+/Y^{3+}$ co-dopant were demonstrated to display efficient PersL (~530 nm)[21]. Given the inter-center transition feature of this unusual luminescence process, we speculated that, with a gradual modification of the crystal environment, D-A could serve as a versatile platform with multicolor fine-tuning capability.

To test our speculation, a series of $Ca_{1-x}Sr_xZnOS:0.1\%Cu^+/1\%Y^{3+}$ ($x = 0-1$) solid solutions were prepared using a solid-state reaction method to accommodate the D-A pairs (Fig. 2a). As depicted in Fig. 2b, X-ray diffraction (XRD) patterns of the as-synthesized samples gradually evolved from CaZnOS (ICSD# 245309) to SrZnOS (ICSD# 431819) with a continuous shift of diffraction peaks towards small angles, suggesting the successful substitution of $Sr^{2+}$ (1.18 Å, CN6) for smaller $Ca^{2+}$ (1.00 Å, CN6) in the host compound. Meanwhile, the associated Rietveld refinement[39] reveals a steady expansion of crystallographic cell volumes with the increase of $Sr^{2+}$ concentration from 0 to 100% (Fig. 2c, Supplementary Fig. 1 and Supplementary Table 2), supporting the formation of single-phase Ca(Sr)ZnOS solid solutions. This result is ascribed to the structural consistency between CaZnOS and SrZnOS, as they share the hexagonal wurtzite structure with a common space group of $P6_3mc$[40]. Notably, the preparation of the intermediate Ca(Sr)ZnOS crystals requires no stringent control of synthetic conditions since they can be formed in a wide temperature range (1173–1373 K) (Supplementary Fig. 2). In contrast, SrZnOS is merely thermodynamically stable at a temperature of ~1348 K (Supplementary Fig. 3), which poses a huge obstacle to the synthesis of pure phase SrZnOS samples (Supplementary Fig. 4)[41,42].

X-ray photoelectron spectroscopy (XPS) was employed to identify the elemental composition of the as-prepared samples, which clearly shows the co-existence of Ca and Sr in a representative sample composed of $Ca_{0.45}Sr_{0.55}ZnOS: Cu^+/Y^{3+}$ (0.1/1%) (Fig. 2d). The Energy-dispersive X-ray analysis (EDAX) further reveals homogeneous distributions of Ca, Sr, Zn, O, and S elements in a single particle level (Fig. 2e and Supplementary Fig. 5). The gradual redshift of the absorption onset in absorption spectra of the $Ca_{1-x}Sr_xZnOS:0.1\%Cu^+/1\%Y^{3+}$ ($x = 0-1$) samples also suggests the continuous modification of the photophysical property along with Sr alloying (Fig. 2f). In short, the formation of pure Ca(Sr)ZnOS compounds with arbitrary Ca to Sr ratios implied the intimate mixing of CaZnOS and SrZnOS at the atomic level, thus providing a large room for manipulating their luminescence properties.

We next characterized the luminescence properties of the Ca(Sr)ZnOS:Cu/Y crystals. We observed that the D-A pairs yield intense photoluminescence (PL) in all Ca(Sr)ZnOS:Cu/Y samples under 365 nm excitation (Fig. 3a). The emission spectra consist of a single broadband with no dependence on the excitation wavelength (Supplementary Fig. 6), attributed to the radiative electron-hole (e–h) recombination from a single type of D-A pair. Intriguing, the PL spectrum shifts to longer wavelengths (523 to 597 nm) along with the change of host composition from CaZnOS to SrZnOS, indicating successful modification of D-A energy states. The PL excitation spectra consist of the host band and characteristic D-A band, both of which shift to longer wavelengths with increasing Sr concentration (Supplementary Fig. 7).

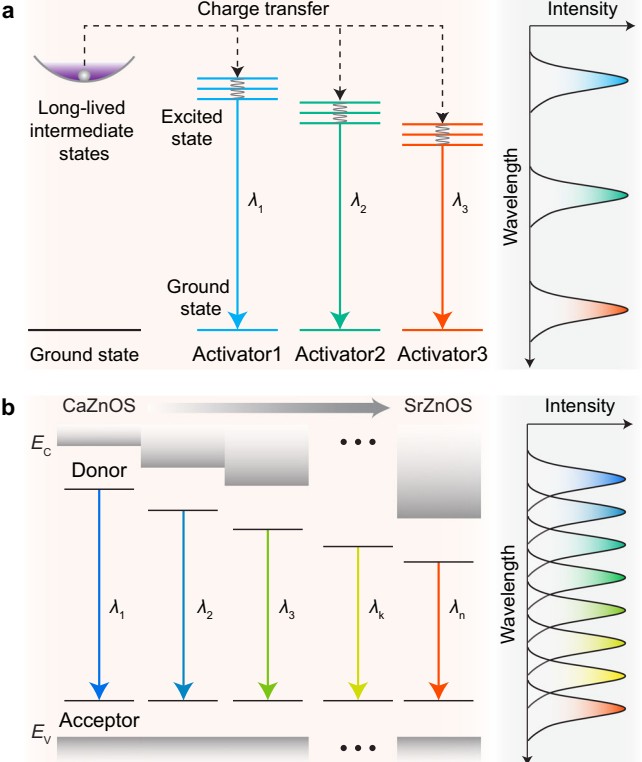

**Fig. 1 | Schematic of the PersL color tuning in various material species.**
**a** Conventional color-tunable PersL emitters offer several discrete emission bands (or the combination thereof). **b** Finely tunable PersL with linearly altered emission peaks can be obtained in isostructural Ca(Sr)ZnOS:D/A (where D stands for the donor and A represents the acceptor) by engineering the D-A emission characteristics. $E_C$ and $E_V$ refer to the energy of conduction band minimum and valence band maximum, respectively.

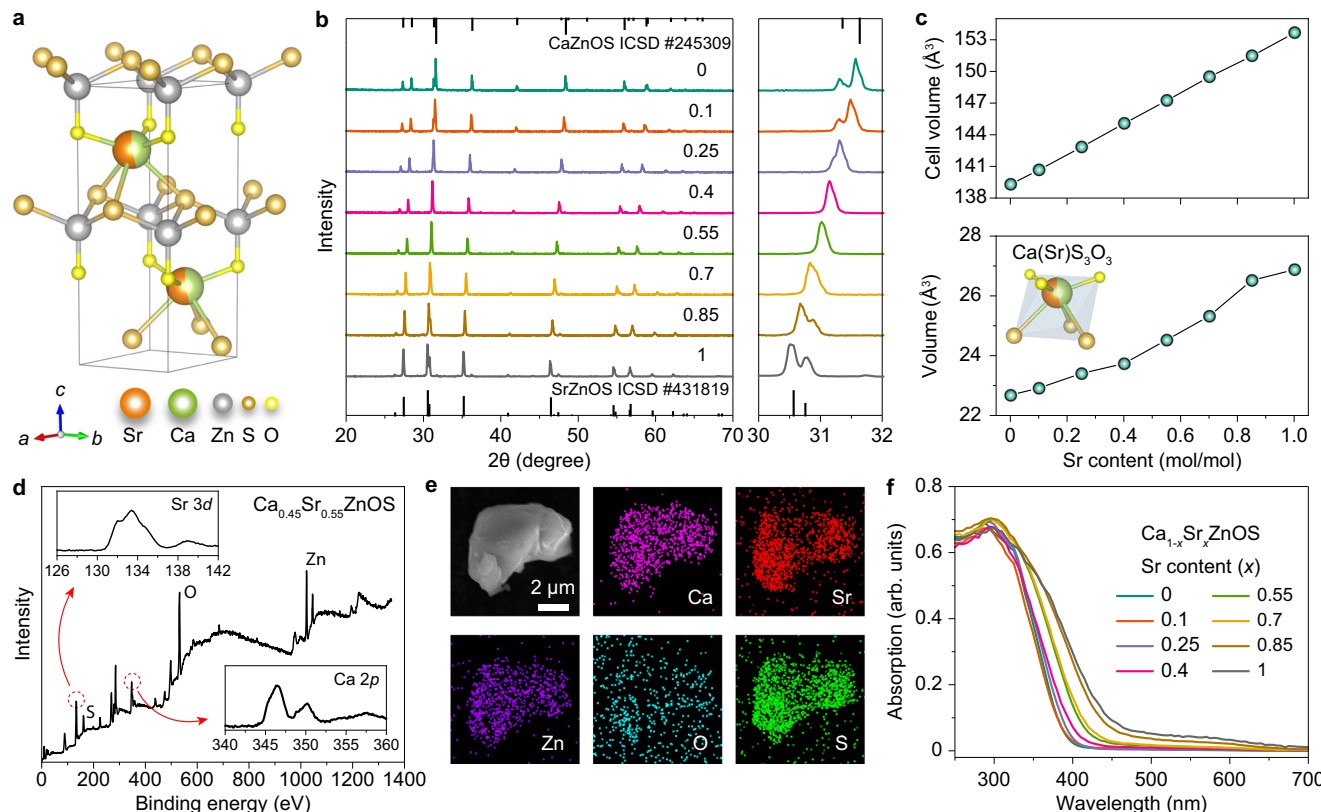

**Fig. 2 | Structural characterization of the alloyed Ca(Sr)ZnOS. a** Schematic presentation of the Ca(Sr)ZnOS crystal structure. **b** Powder XRD patterns (left panel) and enlarged XRD peaks (right panel) of $Ca_{1-x}Sr_xZnOS:0.1\%Cu^+/1\%Y^{3+}$ crystals ($x = 0-1$) sintered at 1348 K for 3 hours. The results show the pure hexagonal phase of the intermediate Ca(Sr)ZnOS solid solutions. The line spectra are the standard diffraction data of CaZnOS and SrZnOS extracted from the Inorganic Crystal Structure Database (ICSD # 245309 and ICSD # 431819), respectively. **c** Rietveld refinement results of the unit cell (top) and $Ca(Sr)S_3O_3$ octahedra (bottom) volumes as a function of Sr content in single-phase Ca(Sr)ZnOS. **d** XPS spectrum of the representative $Ca_{0.45}Sr_{0.55}ZnOS:0.1\%Cu^+/1\%Y^{3+}$ sample. Insets show the high-resolution spectrum of Sr 3$d$ (top) and Ca 2$p$ (bottom) electrons, respectively. **e** EDAX elemental mapping of a single particle of the $Ca_{0.45}Sr_{0.55}ZnOS$ sample. **f** Absorption spectra of the $Ca_{1-x}Sr_xZnOS:0.1\%Cu^+/1\%Y^{3+}$ ($x = 0-1$) crystals. The absorption data were calculated by the Kubelka-Munk equation $\frac{\alpha}{S} = \frac{(1-R)^2}{2R}$, where $\alpha$, $S$, and $R$ represent absorption coefficient, scattering coefficient (assumed to be constant), and absolute reflectance, respectively.

After turning off the irradiation, these crystals display bright PersL (initial brightness up to 5.36 cd m$^{-2}$, Supplementary Table 3) with tunable colors from blue-green to orange (Fig. 3b, c). Particularly, these materials are distinguished by their remarkable ability to be excited by a standard D65 lamp (Supplementary Fig. 8 and Supplementary Table 4), considerably broadening their applicability in lighting, displays, and safety indications. The PersL spectra also comprise a single broadband with a host composition-dependent peak position (527–630 nm), which agrees well with their corresponding steady-state PL emission (Fig. 3d). However, the peak wavelength of PL is observed to be slightly shorter compared to that of PersL across the series of Ca(Sr)ZnOS:Cu/Y crystals. This phenomenon is attributed to the significantly reduced average distance between D-A pairs upon photoexcitation, leading to higher emission energy[21,43]. Significantly, the PersL peak position can be more precisely manipulated by delicately adjusting the Sr$^{2+}$ content (Supplementary Fig. 9), suggesting the high convenience of our strategy for PersL fine-tuning. Furthermore, PersL decay measurement shows that the Ca(Sr)ZnOS:Cu/Y crystals possess a long PersL duration, with a maximum PersL lifetime exceeding 6 h after removing the UV irradiation (Fig. 3e and Supplementary Fig. 10).

Along with the redshifted emission peak, the emission band of PL and PersL are obviously broadened, with the full width at half-maximum (FWHM) varying from 117 to 182 nm for PL and 117 to 170 nm for PersL (Fig. 3d). The difference in the FWHM between PL and PersL originated from the broader distribution of donor-acceptor (D-A) pair

distances in the presence of photoexcitation, which results in a more substantial FWHM for PL when compared to PersL[43]. The pronounced changes in emission peak and FWHM of the D-A pairs indicate that the energy states of D-A pairs are more susceptible to crystal environment than the conventional ionic luminescent centers (e.g., Mn$^{2+}$, Bi$^{3+}$, Pb$^{2+}$, and Ln$^{3+}$) (Supplementary Figs. 11, 12 and Supplementary Table 5). Notably, the emission peaks of PersL can be finely adjusted from 500–550 nm by using Cu$^+$ as the sole dopant, suggesting the presence of D-A states in the Ca(Sr)ZnOS:Cu$^+$ system as well (Supplementary Fig. 13). Taken together, the high spectral tunability of D-A levels allows for the on-demand construction of PersL in an ultra-wide spectral range from 500 to 630 nm.

The luminescence properties of Ca(Sr)ZnOS:Cu/Y significantly exceed the usual expectations for Cu$^{2+}$ ions with a 3$d^9$ electron configuration. If Cu$^{2+}$ ions were involved in the luminescence, emissions or absorptions would likely be in the near-infrared range, similar to the behavior of ZnS:Cu$^{2+}$, instead of the observed visible broadband emission[43]. The luminescence seen in Ca(Sr)ZnOS:Cu/Y is reminiscent of D-A pair emission found in ZnS:Cu$^-$/Al$^{3+}$ (or ZnS:Cu$^+$/Cl$^-$), which suggests that the emission does not stem from isolated ionic emission of Cu$^{2+}$ or Cu$^+$. This observation supports the idea that the luminescence process in Ca(Sr)ZnOS:Cu/Y involves a more complex interaction than simple ionic emission.

To understand how Cu$^+$ and Y$^{3+}$ interact with the Ca(Sr)ZnOS host lattice and produce intense steady-state and long-lasting D-A emission, we performed first-principle calculations based on density functional

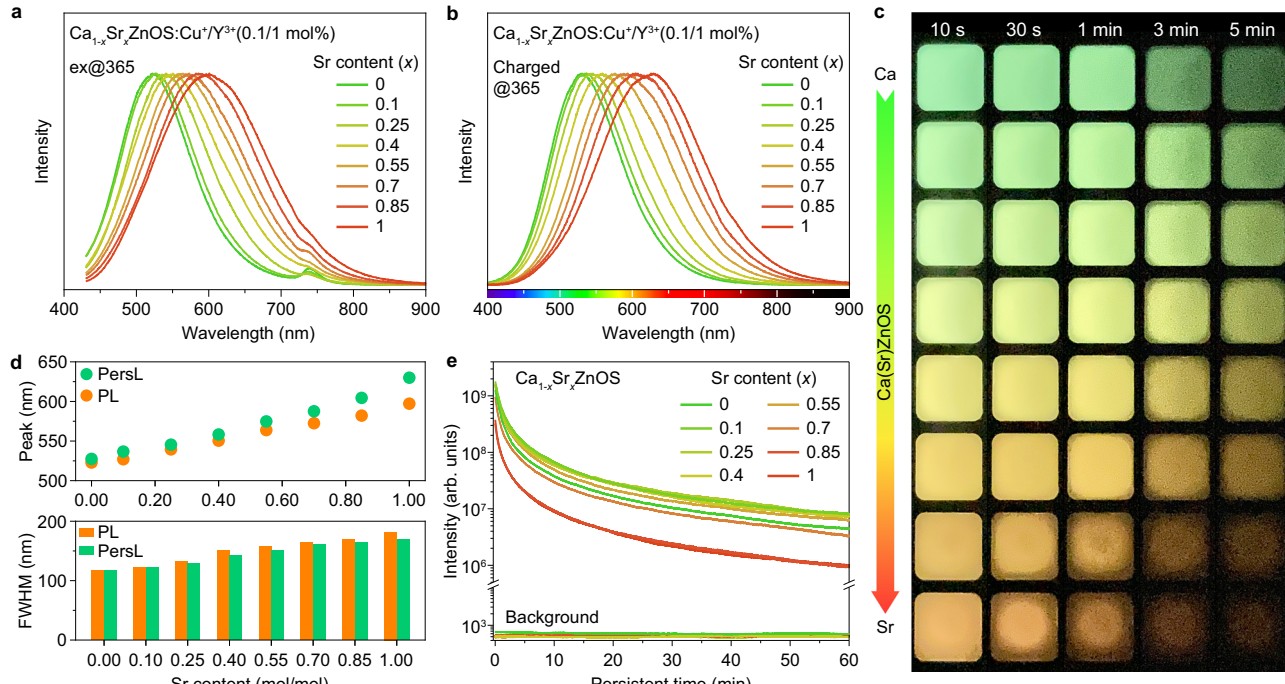

**Fig. 3 | Linear tuning of PersL in Ca(Sr)ZnOS:Cu⁺/Y³⁺. a** PL spectra of Ca₁₋ₓSrₓZnOS:0.1%Cu⁺/1%Y³⁺ ($x = 0$–1) crystals. **b** PersL spectra of Ca₁₋ₓSrₓZnOS:0.1% Cu⁺/1%Y³⁺ ($x = 0$–1) after turning off the UV light (4 W, 365 nm). **c** PersL photographs of Ca(Sr)ZnOS:0.1%Cu⁺/1%Y³⁺ with various Sr contents. **d** Peak wavelength (top) and FWHM (bottom) as a function of Sr concentration in the Ca(Sr)ZnOS:0.1%Cu⁺/1%Y³⁺ crystals. **e** PersL decay profiles of Ca(Sr)ZnOS:0.1%Cu⁺/1%Y³⁺ with various Sr contents. The samples were pre-charged using a 365 nm UV lamp (4 W) for 3 min and a short delay of 20 s was allowed before each measurement.

theory (DFT). Firstly, we obtained a more accurate description of the band edges of CaZnOS:Cu/Y using hybrid DFT. The result revealed a change of the band structure from 4.01 eV (CaZnOS) to 3.33 eV (SrZnOS) with increasing Sr concentration, which is in approximate agreement with our experimental bandgap values (Supplementary Fig. 14 and Supplementary Data 1–4). Then, we calculated the energy level locations of the dopant and possible intrinsic defects of Ca₁₋ₓSrₓZnOS:Cu⁺/Y³⁺ ($x = 0, 0.4, 0.7$, and 1) to understand the charge transition and recombination processes (Fig. 4a). The results suggest that the negative interstitials ($S_i$ and $O_i$) behave like acceptors and the anion vacancies ($V_O$ and $V_S$) act as donors. Meanwhile, the $Cu_{Zn}$ site can also form deep acceptor states to capture excited holes. However, both substitutional defects $Y_{Ca}$ and $Y_{Sr}$ hardly form either donor or acceptor given the distant energy level locations relative to band edges, conforming to our previous observations that different lanthanides generate similar D-A emissions despite the distinguished energy level distributions[21]. By comparing the energy difference of the calculated defect levels between the emission energy of the D-A pairs, we identified that the radiative recombination occurs in $Cu_{Zn}$ acceptors with the electron from $V_O/V_S$ donors (Fig. 4b). In accordance with the simulation, the tunable D-A emission mainly resulted from the steadily lowered donor levels accompanying the deceased bandgap with Sr doping.

To shed more light on the D-A emission behavior, the effect of anion ion was explored by synthesizing a series of Se alloyed Ca₀.₄₅Sr₀.₅₅ZnOS₁₋ₓSeₓ:0.1%Cu⁺/1%Y³⁺ ($x = 0, 0.25, 0.5, 0.75$, and 1) crystals (Supplementary Figs. 15, 16). The prepared Ca(Sr)ZnOS(Se) is highly crystalline and preserves the hexagonal wurtzite structure (Supplementary Fig. 17). Meanwhile, the gradual shifting of absorption onset in UV-vis absorption spectra confirms the successful modification of band structure with Se alloying (Supplementary Fig. 18). The PL and PersL spectra show slightly redshifted D-A emissions (Fig. 4c and Supplementary Fig. 19). However, the Se content affects the emission intensity differently for PL and PersL. The PL intensity is largely

preserved, while the PersL intensity decreases continuously (Fig. 4d and Supplementary Fig. 19). This implies a close association between the trapping state and sulfur. Given the negligible involvement of interstitial S in the trapping/emission mechanisms (Fig. 4b), it can be inferred that sulfur vacancies are the principal determinants of PersL[44]. Our theoretical calculations indicate that selenium vacancies are unlikely to serve as trapping centers, as their energy levels are positioned outside the inter-bandgap, affirming the critical role of $V_S$ in the PersL process (Supplementary Fig. 20 and Supplementary Data 5)[45].

The thermoluminescence (TL) behavior of the Ca(Sr)ZnOS:Cu⁺/Y³⁺ crystals was then studied to gain insight into the trap states. As shown in Fig. 4e, Ca(Sr)ZnOS:Cu⁺/Y³⁺ exhibit nearly identical TL profiles regardless of the Sr content, indicating similar trap distribution among the series of Ca(Sr)ZnOS:Cu⁺/Y³⁺ crystals. However, the maximum of the TL glow peak shifts to a lower temperature (from 359 K to 348 K), indicating a consistent decrease in the trap depth. Additionally, the observed shift in the TL emission peak, in correlation to an increase in Sr content, is consistent with the behavior noted in room-temperature PersL, indicating a similar nature between TL and PersL (Supplementary Fig. 21). The trap depth values can be derived from the TL glow curves by utilizing a variable heating rate method[46]:

$$\frac{\beta E}{kT_m^2} = s e^{-\frac{E}{kT_m}} \tag{1}$$

where $T_m$ is the temperature at maximum glow peak, $\beta$ is the heating rate, $E$ is the trap depth, k is the Boltzmann constant, and $s$ is the frequency factor. Accordingly, we measured the TL spectra for Ca(Sr)ZnOS:Cu/Y crystals over a range of heating rates from 0.5 to 2.5 K s⁻¹ (Supplementary Fig. 22). By plotting $\ln\left(\frac{\beta}{kT_m^2}\right)$ against $\frac{1}{kT_m}$, trap depth can be derived as the absolute value of the slope from the linear fit (Supplementary Fig. 23). As depicted in Fig. 4f, the obtained trap depths are consistent with DFT predictions (Fig. 4b and

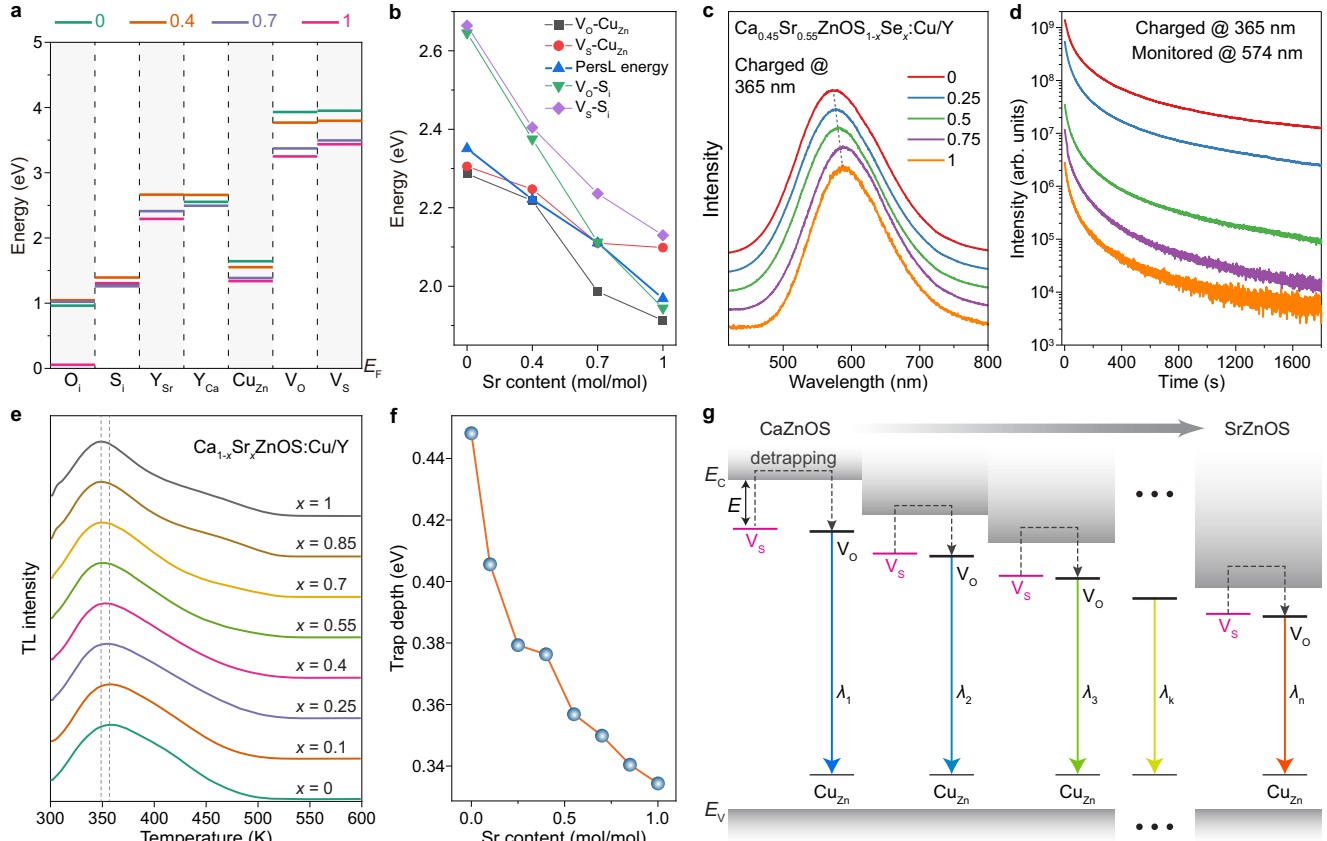

**Fig. 4 | Mechanistic investigations of D-A luminescence characteristics in Ca(Sr)ZnOS:Cu/Y crystals. a** The calculated location of defect states related to the Fermi level in $Ca_{1-x}Sr_xZnOS:Cu^+/Y^{3+}$ ($x = 0$, 0.4, 0.7, and 1). **b** Comparison of the energy difference (between the selected defect levels) and D-A emission energy in Ca(Sr)ZnOS. **c, d** PersL spectra and PersL decay curves of the $Ca_{0.45}Sr_{0.55}ZnOS_{1-x}Se_x:Cu/Y$ ($x = 0$, 0.25, 0.5, 0.75 and 1) crystals. **e** TL spectra of $Ca_{1-x}Sr_xZnOS:0.1\%Cu^+/1\%Y^{3+}$ ($x = 0$, 0.1, 0.25, 0.4, 0.55, 0.7, 0.85 and 1) after charging at 365 nm (4 W). The heating rate is $1\,K\,s^{-1}$. **f** Calculated trap depth as a function of Sr content in the series of Ca(Sr)ZnOS:Cu/Y samples. **g** Band diagram illustration of the PersL mechanism. $E_C$ and $E_V$ refer to the energy of conduction band minimum and valence band maximum, respectively.

Supplementary Fig. 14), showing a decrease from 0.448 to 0.334 eV as the Sr content increases. The continuously decreasing trap depth suggests a more significant reduction in the band edge with respect to trap/donor levels. As a result, a global trap model is utilized[46], conceptualizing trap depth as the relative energy of $V_S$ levels to the conduction band (Fig. 4g).

The availability of the Ca(Sr)ZnOS:Cu⁺/Y³⁺ crystals with high PersL intensity allows us to access intensity-resolvable PersL in the samples for information storage. As proof of concept, we devised a light attenuation mask to control the charging light intensity, which effectively modulates the subsequent PersL brightness (Fig. 5a and Supplementary Fig. 24). Typically, the partially charged sample displayed a lower PersL intensity that turned dark in a shorter time compared to the fully charged counterpart (Fig. 5b and Supplementary Fig. 25). The observations suggested that the trap states of different depths in the sample were populated in a balanced manner at distinct charging intensities, which was confirmed by TL measurements (Fig. 5c and Supplementary Fig. 26).

The photomask-based charging technique can be readily adapted for practical applications (Supplementary Fig. 27). Fig. 5d shows an application of information security by charging a PersL film made of PersL microcrystals and transparent PDMS elastomer through a photomask that is encrypted with a binary code of alphabet letters. The film displays time-evolved information after turning off the charging light (Fig. 5e, f and Supplementary Figs. 28, 29), offering a high level of security for information encryption. By using a photomask with predefined graphics and a flat-panel PersL film, we also establish an

approach for PersL display, which can last 6 hours after UV charging (Supplementary Fig. 10). Due to the continuously tunable emission wavelength, the PersL film can afford a super broadband spectrum based on compositional blending (Fig. 5g), rendering multicolor display through a patterned filter (Fig. 5h). Benefiting from the capability of excitation by the D65 lamp (Supplementary Fig. 30), the multicolor display holds great promise for large-scale billboard applications during nighttime, where a noncontinuous illumination scheme can be adopted, offering an effective solution for energy conservation.

## Discussion

In summary, our investigation of donor/acceptor-activated Ca(Sr)ZnOS crystals highlights an unprecedented opportunity for multicolor PersL tuning. Benefiting from the structural consistency between CaZnOS and SrZnOS, we obtain a vast collection of intermediate $Ca_{1-x}Sr_xZnOS$ ($x = 0–1$) solid solutions with linearly evolved physical properties, which allow us to deliberately control the sub-bandgap D-A levels for PersL fine-tuning. The PersL can be regulated to span a wide wavelength range from 500 to 630 nm in a step down to 5 nm by simply varying the composition of alkaline-earth metals in the host. These findings provide an innovative paradigm for luminescence tuning and may inspire new ideas in designing multicolor materials.

## Methods
### Reagents
CaCO₃ (≥99.0%, Sigma-Aldrich), SrCO₃ (≥99.9%, Sigma-Aldrich), ZnS (99.99%, Sigma-Aldrich), ZnSe (99.99%, Sigma-Aldrich), ZnO (99.99%,

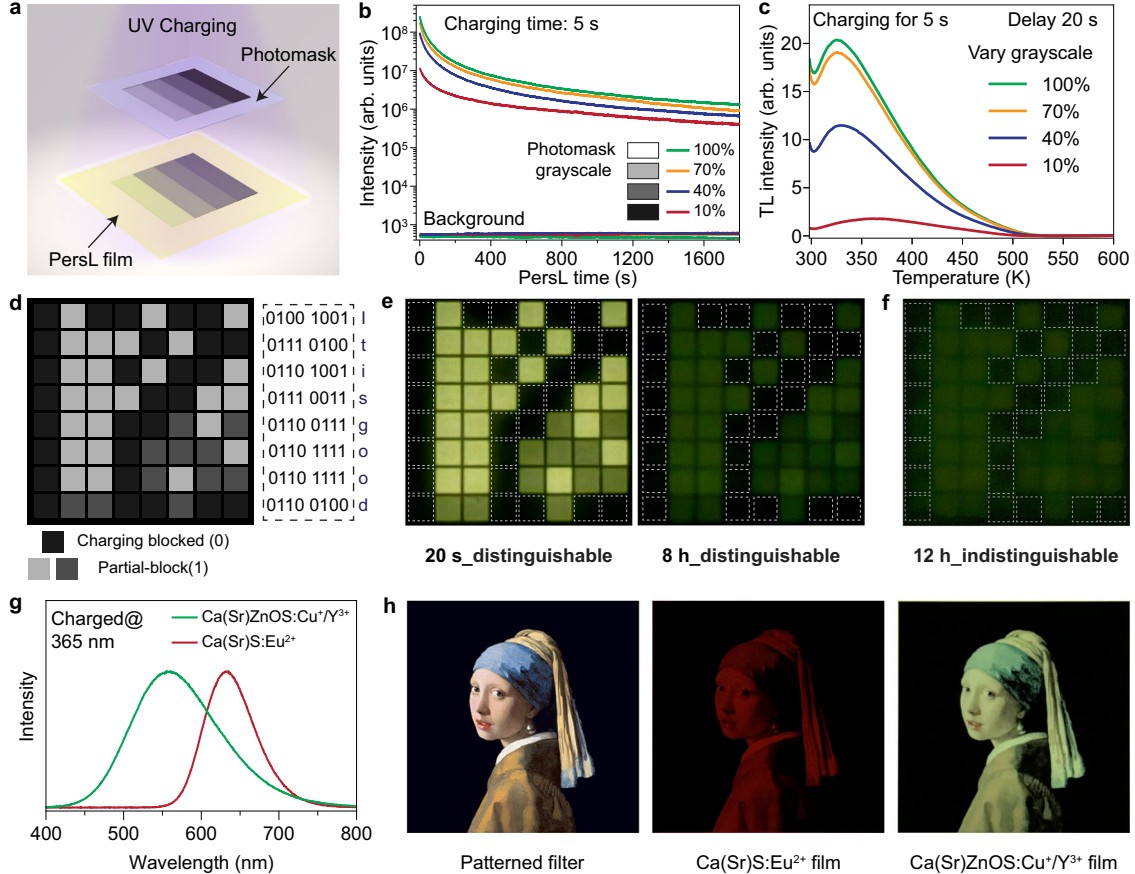

**Fig. 5 | Ca(Sr)ZnOS:Cu/Y PersL crystals for information applications.**
**a** Schematic illustration of photomask charging technique using flat-panel PersL thin film. **b** Comparison of PersL decay of the $Ca_{0.45}Sr_{0.55}ZnOS:0.1\%Cu^+/1\%Y^{3+}$ sample covered by photomasks with various grayscale values during charging (100% and 0% stand for total transmission and depletion of charging light, respectively). **c** TL spectra of the $Ca_{0.45}Sr_{0.55}ZnOS:0.1\%Cu^+/1\%Y^{3+}$ in the presence of various grayscale photomasks during charging. The samples were pre-charged using a 365 nm UV lamp (4 W) for 5 s and a short delay of 20 s was allowed before each measurement. **d–f** Demonstration of programmable information coding based on charging through a patterned photomask. The grayscale values of the

photomask are 10% and 40/70% for charging blocked and partially blocked parts, respectively. The luminescence intensity contrast is used for information encoding. The partially blocked regions (40/70%) remain distinguishable after an 8-hour delay after charging. **g** Super broadband PersL spectrum of blended Ca(Sr)ZnOS:Cu$^+$/Y$^{3+}$ crystals ($Ca_{0.75}Sr_{0.25}ZnOS:Cu^+/Y^{3+}$ and $Ca_{0.45}Sr_{0.55}ZnOS:Cu^+/Y^{3+}$ at a weight ratio of 2:1). The spectrum of red-emitting Ca(Sr)S:Eu$^{2+}$ with narrow band is presented for comparison. **h** Multicolor display through PersL film containing blended Ca(Sr)ZnOS:Cu$^+$/Y$^{3+}$ crystals in the presence of a patterned filter, which shows overwhelming color resolvability compared to the PersL film based on conventional Ca(Sr)S:Eu$^{2+}$.

---

Sigma-Aldrich), CuCl$_2$·2H$_2$O (99%, Alfa Aesar), YF$_3$ (99.9%, Alfa Aesar), Polydimethylsiloxane (PDMS, SYLGARD™ 184 Silicone Elastomer) and its curing agent, were used as received without further processing.

## Preparation of PersL phosphors
All phosphors were synthesized by a high-temperature solid-state reaction method. For the synthesis of $Ca_{1-x}Sr_xZnOS:0.1\%Cu^+/1\%Y^{3+}$ ($x = 0–1$) crystals, stoichiometric raw materials (including CaCO$_3$, SrCO$_3$, ZnS, CuCl$_2$·2H$_2$O and YF$_3$) were weighed, thoroughly mixed, and heat treated in a tube furnace under a constant N$_2$ gas flow (100 mL min$^{-1}$). After natural cooling down to room temperature, the samples were ground to fine powders and stored in a desiccator for further characterization.

The synthesis method of $Ca(Sr)ZnOS_{1-x}Se_x:0.1\%Cu^+/1\%Y^{3+}$ and $Ca(Sr)ZnO_{1+x}S_{1-x}:0.1\%Cu^+/1\%Y^{3+}$ were similar to that of $Ca_{1-x}Sr_xZnOS:0.1\%Cu^+/1\%Y^{3+}$ ($x = 0–1$) except replacing corresponding raw materials.

## Fabrication of flat-panel PersL film
The PersL thin film is composed of a transparent polydimethylsiloxane (PDMS) matrix and micro-sized Ca(Sr)ZnOS:Cu/Y particles. Firstly, SYLGARD™ 184 silicone elastomer base was premixed with the curing

agent at a weight ratio of 10:1. Next, the solution was added with Ca(Sr)ZnOS:0.1%Cu$^+$/1%Y$^{3+}$ crystals under vigorous stirring. The weight ratio of microparticle to PDMS was controlled at 1:2. The slurry was then cast into a 40 mm × 40 mm film (thickness: 200 μm) on a glass substrate by a blade coating method. The as-prepared film was then degassed in a vacuum oven to remove air bubbles and was finally cured at 80 °C for 30 min.

## Details on grayscale patterning
Prior to patterning, the pre-designed grayscale image/binary code was printed onto a transparent polyethylene terephthalate substrate to obtain the photomask. Afterward, the as-prepared photomask was covered on the PersL thin film. After a short period of UV exposure (365 nm, 5–30 s), the grayscale pattern can be directly replicated onto the PersL film. The PersL grayscale image can be directly perceived by the naked eye without further UV exposure.

## Theoretical calculations
All calculations were carried out by using the Vienna Ab Initio Simulation Package. Heyd-Scuseria-Ernzerhof density functionals were applied to guarantee the accuracy of the calculations. The cut-off energy of the plane wave was set as 600 eV throughout the simulation.

Band unfolding was used to calculate the effective band structure of Cu/Y co-doped Ca(Sr)ZnOS. All the structure was optimized until the force between each atom was less than 0.01 eV Å$^{-1}$. The defects equilibrium structure of Cu/Y co-doped Ca(Sr)ZnOS were calculated by using the Perdew-Burke-Ernzerhof (PBE) exchange-correlation function in a 222 supercell, and the defect level is determined by $E_{q_1/q_2} = \frac{E_{q_1}^{\mathrm{f}} - E_{q_2}^{\mathrm{f}}}{q_1 - q_2}$, where $E_{q_1}^{\mathrm{f}}$ and $E_{q_2}^{\mathrm{f}}$ represent the formation energies for the charge states $q_1$, $q_2$, respectively[47–50].

## Physical measurements

XRD patterns were recorded on a Bruker D2 phaser X-ray diffractometer using Cu Kα radiation ($\lambda = 1.5406$ Å). Rietveld refinement of XRD was carried out using the crystallography data analysis software GSAS-II[39]. XPS spectra were measured on a Thermo Scientific ESCALAB Xi⁺ X-ray photoelectron spectrometer microprobe. Scanning electron microscopy (SEM) and EDAX were performed on a JEOL SEM (JSM-IT500). Diffused reflectance spectra were recorded on a Hitachi UH4150 UV-Vis-NIR spectrometer. PL and persistent luminescence spectra were recorded by an Ocean Optics Maya2000 Pro spectrometer in the range of 200–1100 nm. PL excitation spectra were recorded by a Hitachi F-4600 spectrophotometer equipped with an R928 photomultiplier detector. The brightness values of persistent luminescence were measured on a luminance meter (CHROMA METER CS-200). The decay curves of PersL were recorded using the FLS980 spectrometer, set to kinetic scan mode, which tracks the intensity decay at a specific wavelength. In a detailed measurement process, the sample was initially charged with a 365 nm handheld lamp (4 W) for 3 min. Subsequently, the sample was placed inside the spectrometer's chamber. A fixed delay of 20 seconds was allowed between the cessation of the charging and the commencement of the measurement. For luminance calibration, the luminance value at 20 s after ceasing the optical charging was utilized. The calibration assumed that the luminance follows the same trend as the PersL decay curves, and the charging light source and delay time are kept the same for both PersL decay and luminance measurements. Thermoluminescence spectra were recorded by a TL/OSL spectrometer (LTTL-3DS). Prior to measurement, the samples were irradiated by a handheld UV lamp (254 or 365 nm, 4 W) for 3 min. After switching off the light source with a short delay of 1 min, thermoluminescence signals were measured at a constant heating rate. All photographs were obtained using a digital camera (Nikon D5600).

## Data availability

All data supporting the results of this study are available in the Article and its Supplementary Information. Source data are provided with this paper.

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

## Acknowledgements
This work was supported by the Research Grants Council of Hong Kong through a Research Fellowship Scheme (No. RFS2021-1S03) and a General Research Fund (No. 11211922).

## Author contributions
X.Z. and F.W. initiated the project. X.Z., H.S. and F.W. designed the experiments and wrote the paper. X.Z., H.S., J.C., Y.W., X.W., W.Z. and S.L. performed the experiments and analyzed the data. Y.G. conducted the theoretical simulations. X.Z. and S.L. conducted PersL brightness and PersL duration time measurements. All authors contributed to the analysis of this manuscript.

## Competing interests
The authors declare no competing interests.
