## [Peer Review File · Nature Communications]

Continuous Tuning of Persistent Luminescence Wavelength by Intermediate-Phase Engineering in Inorganic CrystalsReviewer #1 (Remarks to the Author):

A long-term goal in the field of persistent luminescence (PersL) is to achieve color-tunable emission in a single material system. Zhang et al. have fulfilled this requirement by manipulating the donor-acceptor emission in the semiconducting Ca(Sr)ZnOS host. For the first time, they demonstrate continuous shifting of the PersL spectrum by simply varying host compositions. The novel PersL materials show remarkable wavelength tunability over a wide spectral range of more than 100 nm with a fine resolution of ~5 nm, which is far beyond the capability of existing PersL material systems. The Ca(Sr)ZnOS PersL phosphors also have high brightness and long PersL lifetime comparable to commercial products. The intermediate-phase engineering approach can provide useful guidance for the future design of PersL materials. Overall, this study represents a breakthrough in PersL research and the manuscript is highly recommended for publication. Some minor suggestions are given below for further improvement of the current work.

1. This manuscript discloses a new PersL spectrum tuning method by engineering the D-A luminescence. I suggest making a comparison table to better demonstrate the advances of current work over previous ones related to PersL tuning.
2. The "Intermediate-Phase Engineering" strategy involves continuous manipulation of the activator energy levels in the lattice-sharing hosts. How can it be applied to other PersL systems?
3. The authors have realized that the intermediate Ca(Sr)ZnOS phases are more accessible than the SrZnOS phase. What is the rationale behind this?
4. The authors should explain the D-A luminescence process in more detail. What was the reason for selecting the Cu/Y pairs as the D-A type? Can the D-A emission be tuned by changing the D-A type in the CaZnOS?
5. The transient luminescence properties of D-A pair should be examined, such as luminescence lifetime and time-resolved emission spectra.

Reviewer #2 (Remarks to the Author):

This paper reports an interesting wavelength tuning method for persistent luminescence (PersL) by intermediate-phase engineering. Using this method, the PersL wavelength could be tailored over a wide range with a high precision. This is a big step forward in terms of wavelength diversification compared to existing material design routes. Although the manuscript is well organized and written in general, there are some points that are unclear and need further polishing. Considering the potential impact in the field optoelectronic materials, I would like to suggest the publication of this paper in Nat. Commun. after appropriate revisions.

1. The authors claim that the emission centers in the Ca(Sr)ZnOS:Y/Cu compounds are donor-accepter (D-A) pairs, which majorly come from the results of first-principle calculations (DFT). Firstly, it should be made clear whether they are donors/acceptors of electrons or holes. Secondly, is it possible that the emission centers are attributed to Cu⁺, Cu²⁺ or other Cu-related defect species, similar to the case of ZnS:Cu.
2. Following the above comment, the previous publications on luminescent properties on Ca(Sr)ZnOS:Cu or its derivatives, if relevant, should be cited and discussed in this paper.
3. The thermoluminescence spectra in Figure 4e contain abnormal spectral signals. Please verify these results.
4. What are the values of trap depths according to Figure 4e or other thermoluminescence glow curves? Are they consistent with the results of DFT calculation?
5. It is not clear enough how to obtain the results of Figure 5h. What is the meaning of patterned filter? Did the author use the synthesized PersL phosphors as lighting sources in the two right panels?
6. I would like to suggest the authors to move Supplementary Figure 18 into the main figures because this is important to understand the charge transfer process in the studied materials.

Reviewer #3 (Remarks to the Author):

The authors report on the luminescence properties of a particular phosphor based on (Ca,Sr)ZnOS:Cu,Y, where tuning of the luminescence (and of the persistent luminescence) can be obtained by changing the Ca-Sr composition. Also a study on the partial substitution of S by Se is

mentioned, but this is less relevant, as an increase in the Se concentration quickly reduces the trapping capacity of the phosphor.

One of the reasons for this study is mentioned in the introduction: PersL emitters of various emission colors usually exhibit unbalanced brightness and efficiency. Unfortunately, this is also the case in the present solution: changing the Ca-Sr ratio also changes the trapping behaviour (as witnessed in the TL glow curves and in the intensity of the decay curves (Fig. 3)). In that sense, the present phosphor system is not a real solution, where spectral tuning can be achieved independently of the trapping behaviour.

Of course, the study by itself contains interesting results, but I am not convinced that the proposed material system will have a high impact in the field, given that the aforementioned problem is not really solved. In the application part, a 'super broadband PersL' is created by mixing two Ca-Sr compositions. Doesn't this create the same issue, regarding color changes during the decay?

Furthermore, in the abstract it is claimed that the high initial brightness is up to 5 cd/m², although I could not find data supporting this claim. All the data for the decay curves are shown in arbitrary units, except for supplementary figure 9, but there the initial intensity is lower than 5cd/m². Furthermore, it is mentioned that it was measured with a (scanning) spectrometer, calibrated with a luminance meter. More details should definitely be provided on the calibration method, as the luminance (measured in cd/m²) depends on the shape of the spectrum. Hence, a calibration of the FLS980 spectrometer to yield cd/m² values for widely different emission colors is not straightforward. The advise would be to measure the phosphors directly, after excitation, with the luminance meter, and then compare them. Preferably, also standard excitation conditions (Xe lamp and D65 lamp) should be used to give a fair estimate of the obtainable brightness.

It would also be interesting to plot in Fig. 3d the peak and fwhm not only for the PL, but also for the PersL, and discuss in more detail the differences.

There is an issue with the TL curves in Figure 4. The erratic way in which the plot lines are shown does not have a meaning (a TL measurement assumes a constant heating rate, now the curves jump back and forth as a function of temperature). Also, it is mentioned that the TL glow curves are nearly identical, but the differences are actually quite large and they do not change in a very systematic way when the composition is changed. For a publication in NatComm, this should be investigated in more detail.

I will not comment on the theoretical calculations, as I am not an expert in the field. Regarding the conclusion from the Se substitution, it is a bit surprising to read that the PersL is strongly correlated with sulfur, while it is actually a vacancy. This should be explained in more detail.

Finally, I must admit that – even after reading several times - I didn't get how the programmable optical storage would actually be used in practice. In panel f, other information appears compared to panel e (Figure 5), but how does this change evolve over time, and what is then the "useful" time period to do the read out? I do not see where an optical information storage during 5s (or 60s) would be useful for. Also for the multicolor display, there is some uncertainty about what is exactly done. Was the phosphor excited through the patterned film? Or would such a 'display' be excited without the pattern, after which the user inserts the film? Furthermore, is the emission spectra stable through the mentioned duration of 6 hours? Is it then still possible to see color information?

Point-by-point response to reviewers' comments

We acknowledge the reviewers for taking the time to review our manuscript and providing constructive comments. We have revised our manuscript to address all the questions or comments. The changes are marked in **red** in the revised manuscript. Below are our detailed responses to the reviewers' comments (in black) and the actions taken (in **blue**).

Reviewer #1 (Remarks to the Author):

A long-term goal in the field of persistent luminescence (PersL) is to achieve color-tunable emission in a single material system. Zhang et al. have fulfilled this requirement by manipulating the donor-acceptor emission in the semiconducting Ca(Sr)ZnOS host. For the first time, they demonstrate continuous shifting of the PersL spectrum by simply varying host compositions. The novel PersL materials show remarkable wavelength tunability over a wide spectral range of more than 100 nm with a fine resolution of ~5 nm, which is far beyond the capability of existing PersL material systems. The Ca(Sr)ZnOS PersL phosphors also have high brightness and long PersL lifetime comparable to commercial products. The intermediate-phase engineering approach can provide useful guidance for the future design of PersL materials. Overall, this study represents a breakthrough in PersL research and the manuscript is highly recommended for publication. Some minor suggestions are given below for further improvement of the current work.

Response: We appreciate the recommendation and your recognition of the strengths of our study.

1. This manuscript discloses a new PersL spectrum tuning method by engineering the D-A luminescence. I suggest making a comparison table to better demonstrate the advances of current work over previous ones related to PersL tuning.

Response: We are grateful for your valuable suggestion. The latest advancements in PersL spectrum tuning are concisely documented in **Supplementary Table 1** in the revised Supplementary Information. The prevalent method for PersL color modulation entails the amalgamation of diverse emission centers, including triplet states, guest chromophores, and inorganic activators, into a singular host matrix. These strategies yield specific emission bands yet impede the comprehensive full-spectrum expression of PersL.

Supplementary Table 1. Brief summary of literature work on PersL spectrum tuning.

Materials	Emissions	Spectrum tunability	Ref.
Polycyclic aromatic hydrocarbons (PAHs)-melamine formaldehyde (MF) composites	466 nm (TP-MF) 503 nm (PA-MF) 544 nm (FA-MF) 592 nm (Py-MF) 601 nm (BA-MF)	Discrete	Angew. Chem. Int. Ed. 2024, 136, e202318516
Fluorescein sodium (FluNa)-Al ₂ (SO ₄) ₃	483, 549 nm	Discrete	Angew. Chem. Int. Ed. 2023, 135, e202217616
Calcein sodium (CalNa)-Al ₂ (SO ₄) ₃	466, 548 nm	Discrete	Angew. Chem. Int. Ed. 2023, 135, e202217616
MCATMA/C545T	408, 526, 560, 580 and 600 nm	Discrete	Adv. Mater. 2022, 34, 2206712
Cytosine-Cd/ZnX ₂ (X = Cl, Br)	435, 513 nm (Cy-CdCl ₂) 435, 513 nm (Cy-ZnCl ₂)	Discrete	Adv. Sci. 2022, 9, 2200992

Poly(acrylamide-co-N-vinylcarbazole) based host-guest composites	414 nm (Host) 502 nm (PBD, guest) 560 nm (Fluc, guest) 570 nm (Rh123, guest) 620 nm (RhB, guest)	Discrete	Sci. Adv. 2022, 8, eabk2925
(R,R)-DAACH/(S, S)-DAACH	470–530 nm	Discrete (Excitation dependent)	Nat. Commun. 2022, 13, 429
4,4'-bis(N-carbazolyl)-1,1'-biphenyl (pCBP)	430, 560 nm	Discrete	Angew. Chem. Int. Ed. 2020, 59, 10032
2,4,6-trimethoxy-1,3,5-triazine (TMOT)	465–505 nm	Discrete (Excitation dependent)	Nat. Photonics 2019, 13, 406
Organic molecules (H-aggregates)	530, 575 nm (DPhCzT) 515, 547 nm (DEOPh) 529, 574 nm (DECzT) 543, 591 nm (CzDCIT) 587, 644 nm (DCzPhP)	Discrete	Nat. Mater. 2015, 14, 685
Multicomponent copolymer	445, 517 nm (PDNA) 445, 514 nm (PDBA) 489, 546 and 584 nm (Tb ³⁺) 385 nm (Nd ³⁺) 453 nm (Tm ³⁺) 542 nm (Ho ³⁺) 543 nm (Er ³⁺) 573 nm (Dy ³⁺) 594 nm (Sm ³⁺) 606 nm (Pr ³⁺)	Discrete	Nat. Commun. 2020, 11, 944
NaLuF ₄ :Ln ³⁺ /Gd ³⁺ (Ln ³⁺ : trivalent lanthanide ion)	UVC (Sr ₂ Al ₂ SiO ₇) 266 nm (Sr ₃ Y ₂ Si ₆ O ₁₈) 268 nm (Ca ₂ Al ₂ SiO ₇) 267 nm (Ca ₃ Al ₂ Si ₃ O ₁₂) 270 nm (Lu ₂ SiO ₅) 267 nm (LiYSiO ₄)	Discrete	Nature 2021, 590, 410
Pr ³⁺ doped in multiple hosts	267 nm (LiYSiO ₄)	Discrete	Nat. Commun. 2020, 11, 2040
LiGa ₅ O ₈ :Mn ²⁺ transparent glass ceramic	510, 625 nm	Discrete	Light Sci. Appl. 2020, 9, 22
CdSiO ₃ @SiO ₂ nanoparticles	438 nm (In ³⁺) 438, 580 nm (In ³⁺ and Mn ²⁺) 549 nm (Tb ³⁺) 578 nm (Dy ³⁺) 620 nm	Discrete	Adv. Mater. 2020, 32, 2003881
SrSi ₂ O ₂ N ₂ :Ln ²⁺ /Ln ³⁺ (Ln: lanthanides)	(Yb ²⁺ /Dy ³⁺ , Yb ²⁺ /Ho ³⁺ , Yb ²⁺ /Er ³⁺) 540 nm	Discrete	ACS Appl. Mater. Interfaces 2018, 10, 1854
NaYF ₄ :Ln ³⁺ @ NaYF ₄ (Ln: lanthanides)	(Eu ²⁺ /Dy ³⁺ , Eu ²⁺ /Ho ³⁺ , Eu ²⁺ /Er ³⁺) 1525 nm (Er) 1475 nm (Tm) 1180 nm (Ho) 1064 nm (Nd)	Discrete	Nat. Nanotechnol. 2021, 16, 1011
Cs ₂ CdCl ₄ and Cs ₂ CdCl ₄ :Mn ²⁺	500, 612 nm (Cs ₂ CdCl ₄) 600 nm (Cs ₂ CdCl ₄ :Mn ²⁺)	Discrete	Angew. Chem. Int. Ed. 2023, 62, e202308420
CaZnOS:Pb ²⁺ /Ln ³⁺ and CaZnOS:Cu ⁺ /Ln ³⁺	Multiple emission centers	Discrete	Laser Photonics Rev. 2023, 17, 2300132
CaAl ₂ O ₄ :Eu ²⁺ Nd ³⁺ /CsPbX ₃ /PDMS composite (X: Cl ⁻ , Br ⁻ , I ⁻)	440–694 nm (quantum dots as color converters)	Continuous (complex composite)	Angew. Chem. Int. Ed. 2019, 131, 7017

2. The "Intermediate-Phase Engineering" strategy involves continuous manipulation of the activator energy levels in the lattice-sharing hosts. How can it be applied to other PersL systems?

Response: The concept of "Intermediate-Phase Engineering" offers a novel approach to control the emission characteristics of luminescent materials. This technique particularly benefits ions that are highly sensitive to changes in the crystal field environment, such as Ce^{3+} , Eu^{2+} , and Cr^{3+} . When embedded in a suitable host lattice, these ions serve as localized luminescence centers whose emission can be finely tuned. Our previous research has demonstrated this capability, where we successfully manipulated the **mechanoluminescence** properties of $\text{Ga}_2\text{O}_3:\text{Cr}^{3+}$ by altering the crystal field strength. This was detailed in our publication (*Matter* **2023**, *6*, 2935), highlighting the potential of this method for precise control over the emission spectrum. Building upon this foundation, we propose to use Cr^{3+} -based PersL materials as a starting point for further research. By initiating with a Cr^{3+} enriched PersL material, we anticipate the possibility of continuously adjusting the near-infrared (NIR) PersL emission spectrum through intermediate-phase engineering.

3. The authors have realized that the intermediate $\text{Ca}(\text{Sr})\text{ZnOS}$ phases are more accessible than the SrZnOS phase. What is the rationale behind this?

Response: We are grateful for the question posed. It has been reported previously that the synthesis of SrZnOS was not achievable (*Inorganic Chemistry* **2007**, *46*, 2571), and further research highlighted the essentiality of stringent control over the reaction conditions as well as the water sonification cleaning process (*J. Solid State Chem.* **2017**, *246*, 225), underlining the replication difficulties of the compound. Our findings align with prior research, indicating that SrZnOS forms within a very narrow temperature range, specifically around 1075 °C (This specific temperature corresponds to 1050 °C in the reference work). This limited temperature window renders the synthesis conditions highly critical. Additionally, the compound's inherent crystal structure instability could result in structural distortion beyond these critical temperatures. Notably, our XRD measurements (as shown in **Supplementary Fig. 3**) reveal that at temperatures either lower or higher than 1075 °C (1348 K), impurities such as SrS and ZnO emerge. These impurities likely arise from the decomposition of the target SrZnOS , given that ZnS and SrO (SrCO_3) were utilized as source materials. For the intermediate compound $\text{Ca}(\text{Sr})\text{ZnOS}$ or CaZnOS , we successfully obtained pure phases within the temperature range of 900–1100 °C (1173–1373 K), suggesting their greater crystal structure stability compared to SrZnOS .

4. The authors should explain the D-A luminescence process in more detail. What was the reason for selecting the Cu/Y pairs as the D-A type? Can the D-A emission be tuned by changing the D-A type in the CaZnOS ?

Response: Thank you for raising this point. Donor-acceptor (D-A) luminescence originates from the recombination of an electron bound to a donor with a hole bound to an acceptor, both of which are defect levels within the semiconductor's bandgap. Our previous research revealed that Cu/RE ($\text{RE} = \text{Y}, \text{Gd}, \text{Tb}, \text{Nd}, \text{Er}, \text{Ho}, \text{Tm}, \text{Dy}, \text{and Pr}$) can form D-A pairs, leading to band emissions in addition to the characteristic luminescence from the RE ions in CaZnOS . Consequently, in the current study, we have employed the non-luminescent Y as a co-activator to exclusively generate the D-A emission and modulate the persistent luminescence (PersL) spectrum.

The D-A emission's tunability was demonstrated by substituting Cu/Y with other D-A pairs. For instance, using Cu/Gd resulted in a continuously adjustable D-A spectrum, comparable to Cu/Y (**Supplementary Fig. 12a-b**). With Cu/Tb , the resulting spectrum was a composite of D-A and Tb ionic emissions, with the D-A luminescence being constantly adjusted with increasing

Sr concentration (**Supplementary Fig. 12c-d**). Additionally, we observed emissions in Ag/In co-doped Ca(Sr)ZnOS, in which the emission peaks shifted to a shorter wavelength as the Sr content increased, despite without a detectable PersL signal (**Supplementary Fig. 12e**).

We have added the following discussion in the revised manuscript:

"Concurrently, our prior investigations have demonstrated that, in CaZnOS, copper and rare-earth elements (e.g., Y, Gd, Tb, Nd, Er, Ho, Tm, Dy and Pr) are capable of forming D-A pairs. This interaction facilitates band emissions, which occur alongside the intrinsic lanthanide luminescence. Consequently, in the current study, we employed Cu and non-luminescent Y as dopants to exclusively generate D-A emission for PersL modulation."

Supplementary Fig. 12. Tunable D-A luminescence in Ca(Sr)ZnOS crystals doped with various activators: a-b) Cu/Gd, c-d) Cu/Tb, e) Ag/In.

5. The transient luminescence properties of D-A pair should be examined, such as luminescence lifetime and time-resolved emission spectra.

Response: We appreciate your suggestion. The radiative recombination rate of donor-acceptor (D-A) pairs is influenced by the separation distance between them (*Phys. Rev.* **1965**, 140, A202). The distribution of D-A separations typically results in a non-exponential photoluminescence

(PL) decay signature for D-A luminescence. In our study, however, the PL decay measurements were significantly affected by a strong persistent luminescence (PersL) signal. We observed a rapid decay phase followed by a slower decay in the representative $\text{Ca}_{0.45}\text{Sr}_{0.55}\text{ZnOS}:\text{Cu}/\text{Y}$ sample, which is indicative of this interference. Additionally, the transient emission spectra are also included, confirming the presence of the PersL signal.

Fig. R1. a) Photoluminescence decay and b) transient emission spectra of the representative $\text{Ca}_{0.45}\text{Sr}_{0.55}\text{ZnOS}:\text{Cu}/\text{Y}$ sample.

Reviewer #2 (Remarks to the Author):

This paper reports an interesting wavelength tuning method for persistent luminescence (PersL) by intermediate-phase engineering. Using this method, the PersL wavelength could be tailored over a wide range with a high precision. This is a big step forward in terms of wavelength diversification compared to existing material design routes. Although the manuscript is well organized and written in general, there are some points that are unclear and need further polishing. Considering the potential impact in the field optoelectronic materials, I would like to suggest the publication of this paper in Nat. Commun. after appropriate revisions.

Response: Thank you for your recognition and positive comments.

1. The authors claim that the emission centers in the Ca(Sr)ZnOS:Y/Cu compounds are donor-accepter (D-A) pairs, which majorly come from the results of first-principle calculations (DFT). Firstly, it should be made clear whether they are donors/acceptors of electrons or holes. Secondly, is it possible that the emission centers are attributed to Cu⁺, Cu²⁺ or other Cu-related defect species, similar to the case of ZnS:Cu.

Response: We apologize for any confusion. In semiconducting materials, donor impurities donate additional electrons, whereas acceptor impurities are responsible for generating free holes. Radiative recombination is initiated when the wavefunction of an electron confined within a donor site overlaps with that of a hole in an acceptor, leading to donor-acceptor (D-A) luminescence. To verify the D-A emission nature, we have checked the peak and FWHM for both PL and PersL (Fig. 3d). The peak wavelength of PL is shorter (corresponding to higher emission energy) than that of PersL for the series of CaZnOS:Cu/Y crystals and the FWHM values of PL are larger than the correspondent PersL. The difference can be understood by considering the transition energy of D-A pair luminescence, which can be expressed as:

$$E = E_g - (E_D + E_A) + \frac{e^2}{4\pi\epsilon r}$$

where E_g is the bandgap energy, E_D/E_A is the ionization energy of the donor/acceptor, ϵ is the dielectric constant of the crystal and r is the distance between D-A pair. During photoexcitation, the average distance between donor-acceptor pairs significantly decreases. This reduction in distance results in an increase in recombination energy E , which consequently leads to a decrease in the emission wavelength when compared to that of PersL. In a similar situation, photoexcitation is responsible for a wider distribution of donor-acceptor pair distances, which manifests as a broader band shape relative to PersL. Furthermore, our previous research has established that D-A emission is directly proportional to the excitation intensity, where higher excitation power results in a reduced separation distance between donor and acceptor, thereby inversely affecting the recombination energy (*Laser Photonics Rev.* **2023**, *17*, 2300132). Such D-A luminescence mirrors the interactions between the activator (typically Cu⁺) and the co-activator (such as Cl⁻ or Al³⁺), as seen in the luminescence dynamics of ZnS-based electroluminescent or PersL materials (*J. Electrochem. Soc.* **1956**, *103*, 342; *J. Electrochem. Soc.* **1953**, *100*, 72).

We have incorporated a discussion on the characteristics of D-A emission in both PL and PersL into the revised manuscript:

"However, the peak wavelength of PL is observed to be slightly shorter compared to that of PersL across the series of Ca(Sr)ZnOS:Cu/Y crystals. This phenomenon is attributed to the significantly reduced average distance between D-A pairs upon photoexcitation, leading to higher emission energy.

Along with the redshifted emission peak, the emission band of PL and PersL are obviously broadened, with the full width at half-maximum (FWHM) varying from 117 to 182 nm for PL and 117 to 170 for PersL (Fig. 3d). The difference in the FWHM between PL and PersL originated from the broader distribution of donor-acceptor (D-A) pair distances in the presence of photoexcitation, which results in a more substantial FWHM for PL when compared to PersL."

Fig. 3. Linear tuning of PersL in Ca(Sr)ZnOS:Cu⁺/Y³⁺. **a**, PL spectra of Ca_{1-x}Sr_xZnOS:0.1%Cu⁺/1%Y³⁺ (x = 0-1) crystals. **b**, PersL spectra of Ca_{1-x}Sr_xZnOS:0.1%Cu⁺/1%Y³⁺ (x = 0-1) after turning off the UV light (4 W, 365 nm). **c**, PersL photographs of Ca(Sr)ZnOS:0.1%Cu⁺/1%Y³⁺ with various Sr contents. **d**, PL/PersL peak wavelength (top) and FWHM (bottom) as a function of Sr concentration in the Ca(Sr)ZnOS:0.1%Cu⁺/1%Y³⁺ crystals. **e**, PersL decay profiles of Ca(Sr)ZnOS:0.1%Cu⁺/1%Y³⁺ with various Sr contents. The samples were pre-charged using a 365 nm UV lamp (4 W) for 3 min and a short delay of 20 s was allowed before each measurement.

The luminescence properties of Ca(Sr)ZnOS:Y/Cu are notably superior to what is typically expected from Cu²⁺ ions with a 3d⁹ electron configuration. Should Cu²⁺ ions play a role in the luminescence process, one would expect to observe emissions or absorptions in the near-infrared spectrum, similar to the behaviour of ZnS:Cu²⁺, rather than a visible broadband emission. Regarding the possibility of Cu-related defect luminescence, the spectral analysis of Ca(Sr)ZnOS:Cu samples has been presented. These specimens demonstrate donor-acceptor (D-A) features, with an emission peak observed between 500-550 nm, indicative of donor sites potentially originating from Cu-induced charge-compensating defects.

We have added the following discussion in the revised manuscript:

"The luminescence properties of Ca(Sr)ZnOS:Cu/Y significantly exceed the usual expectations for Cu²⁺ ions with a 3d⁹ electron configuration. If Cu²⁺ ions were involved in the luminescence, emissions or absorptions would likely be in the near-infrared range, similar to the behaviour of ZnS:Cu²⁺, instead of the observed visible broadband emission. The luminescence seen in Ca(Sr)ZnOS:Cu/Y is reminiscent of D-A pair emission found in

ZnS:Cu⁻/Al³⁺ (or ZnS:Cu⁺/Cl⁻), which suggests that the emission does not stem from isolated ionic emission of Cu²⁺ or Cu⁺. This observation supports the idea that the luminescence process in Ca(Sr)ZnOS:Cu/Y involves a more complex interaction than simple ionic emission."

2. Following the above comment, the previous publications on luminescent properties on Ca(Sr)ZnOS:Cu or its derivatives, if relevant, should be cited and discussed in this paper.

Response: We appreciate your suggestion. Tu et al. explored the mechanical quenching of PersL in CaZnOS:Cu⁺ (*Light Sci. Appl.* **2015**, 4, e356; *Appl. Phys. Lett.* **2014**, 105, 011908). They concluded that the luminescence originates from a donor-acceptor (D-A) pair, formed by interactions between nearby trap levels and the Cu²⁺ (photoionized Cu⁺) energy state. Our previous studies have indicated that the introduction of RE³⁺ (where RE represents rare earth elements) modifies the donor-acceptor (D-A) emission spectrum, which can be further finely tuned by Sr²⁺ alloying, as detailed in our work.

We have cite related reference (ref. 37–38) and added the following discussion in the revised manuscript:

"The study by Tu et al. has unveiled D-A emission characteristics through their research on the mechanical quenching of PersL in CaZnOS:Cu⁺. Concurrently, our prior investigations have demonstrated that in CaZnOS, copper and rare earth elements (e.g., Y, Gd, Tb, Nd, Er, Ho, Tm, Dy and Pr) are capable of forming donor-acceptor pairs. This interaction facilitates band emissions, which occur alongside the intrinsic lanthanide luminescence."

3. The thermoluminescence spectra in Figure 4e contain abnormal spectral signals. Please verify these results.

Response: We apologize for any confusion. The TL measurement instrument has been upgraded to a domestic model, denoted as TL/OSL spectrometer (LTTL-3DS). This advanced device is capable of resolving wavelength information, which adds an additional dimension to our understanding of the TL signal. By integrating across wavelengths, we can derive the conventional two-dimensional TL glow curve, which depicts TL intensity as a function of temperature. Moreover, by rigorously controlling the testing conditions (pre-charging wavelength/time, delay time, heating rate), we have determined that **the profile of TL glow curves kept nearly unchanged** while their **peak maximums shift slightly to the lower temperature side** (from 359 K to 348 K at a constant heating rate of 1 K/s) as the Sr concentration increases. This finding substantiates our hypothesis that alloying with Sr does not significantly alter the trap distribution; nonetheless, it consistently results in a decrease in the values of trap depth. For additional information, please refer to the detailed explanation in response to *Question 4*. The result has been incorporated into **Fig. 4e** in the revised manuscript. In addition, the original three-dimensional contour plots of TL spectra have been included in **Supplementary Fig. 21**. Owing to their consistent emission characteristics, TL also exhibits a spectral profile that can be finely adjusted, similar to that of room temperature PersL.

We have added the following discussion in the revised manuscript:

"However, the maximum of the TL glow peak shifts to a lower temperature (from 359 K to 348 K), indicating a consistent decrease in the trap depth. Additionally, the observed shift in the TL emission peak, in correlation to an increase in Sr content, is consistent with the behaviour noted in room-temperature PersL, indicating a similar nature between TL and PersL (**Supplementary Fig. 21**)."

Fig. 4. Mechanistic investigations of D-A luminescence characteristics in Ca(Sr)ZnOS:Cu/Y crystals. **a**, The calculated location of defect states related to the Fermi level in $\text{Ca}_{1-x}\text{Sr}_x\text{ZnOS}:\text{Cu}^+/\text{Y}^{3+}$ ($x = 0, 0.4, 0.7$ and 1). **b**, Comparison of the energy difference (between the selected defect levels) and D-A emission energy in Ca(Sr)ZnOS. **c-d**, PersL spectra and PersL decay curves of the $\text{Ca}_{0.45}\text{Sr}_{0.55}\text{ZnOS}_{1-x}\text{Se}_x:\text{Cu}/\text{Y}$ ($x = 0, 0.25, 0.5, 0.75$ and 1) crystals. **e**, TL spectra of $\text{Ca}_{1-x}\text{Sr}_x\text{ZnOS}:0.1\%\text{Cu}^+/1\%\text{Y}^{3+}$ ($x = 0, 0.1, 0.25, 0.4, 0.55, 0.7, 0.85$ and 1) after charging at 365 nm (4 W). The heating rate is 1 K/s. **f**, Calculated trap depth as a function of Sr content in the series of Ca(Sr)ZnOS:Cu/Y samples. **g**, Band diagram illustration of the PersL mechanism.

Supplementary Fig. 21. Contour mapping of TL intensity as a function of emission wavelength and temperature for $\text{Ca}_{1-x}\text{Sr}_x\text{ZnOS:0.1\%Cu}^+/1\%\text{Y}^{3+}$ ($x = 0, 0.1, 0.25, 0.4, 0.55, 0.7, 0.85$ and 1). The heating rate during TL measurement is 1 K/s . The shift in the emission peak with increasing Sr content aligns with the behaviour observed in room temperature PersL, indicating a similar luminescence nature between TL and PersL.

We have also updated the TL results for photomask charging and variable delay time, utilizing the newly improved TL measurement model.

Supplementary Fig. 26. a) TL spectra of $\text{Ca}_{0.45}\text{Sr}_{0.55}\text{ZnOS:0.1\%Cu}^+/\text{1\%Y}^{3+}$ in the presence of various grayscale masks during charging. The samples were pre-charged using a 365 nm UV lamp (4 W) for 5 s, and a short delay of 20 s was allowed before each measurement. b) TL spectra of $\text{Ca}_{0.45}\text{Sr}_{0.55}\text{ZnOS:0.1\%Cu}^+/\text{1\%Y}^{3+}$ (Charged for 5 s; 100% grayscale photomask) with various delay times before each measurement.

4. What are the values of trap depths according to Figure 4e or other thermoluminescence glow curves? Are they consistent with the results of DFT calculation?

Response: Thank you for raising this question. We investigate the evolution of trap depth using variable heating rate TL measurements. The resulting trap depth exhibits a decreasing trend with increasing Sr concentration. We have incorporated the results into **Fig. 4f** and included additional discussion in the revised manuscript:

"The trap depth values can be derived from the TL glow curves by utilizing a variable heating rate method:

$$\frac{\beta E}{kT_m^2} = s e^{-\frac{E}{kT_m}}$$

where T_m is the temperature at maximum glow peak, β is the heating rate, E is the trap depth, k is the Boltzmann constant and s is the frequency factor. Accordingly, we measured the TL spectra for $\text{Ca}(\text{Sr})\text{ZnOS:Cu/Y}$ crystals over a range of heating rates from 0.5 to 2.5 K/s (**Supplementary Fig. 22**). By plotting $\ln\left(\frac{\beta}{kT_m^2}\right)$ against $\frac{1}{kT_m}$, trap depth can be derived as the absolute value of the slope from the linear fit (**Supplementary Fig. 23**). As depicted in **Fig. 4f**, the obtained trap depths are consistent with DFT predictions (**Fig. 4b and Supplementary Fig. 14**), showing a decrease from 0.448 to 0.334 eV as the Sr content increases."

Supplementary Fig. 22. TL spectra of $\text{Ca}_{1-x}\text{Sr}_x\text{ZnOS:0.1\%Cu}^+/1\%\text{Y}^{3+}$ ($x = 0, 0.1, 0.25, 0.4, 0.55, 0.7, 0.85$ and 1) recorded at different heating rates. The samples were pre-charged using a 365 nm UV source (4 W) for 180 s, and a constant delay of 60 s was allowed before each measurement. The heating rate was varied from 0.5 to 2.5 K/s.

Supplementary Fig. 23. $\ln\left(\frac{\beta}{kT_m^2}\right)$ versus $\frac{1}{kT_m}$ plots as derived from the TL curves of $\text{Ca}_{1-x}\text{Sr}_x\text{ZnOS:0.1\%Cu}^+/1\%\text{Y}^{3+}$ ($x = 0, 0.1, 0.25, 0.4, 0.55, 0.7, 0.85$ and 1). The heating rate β was varied from 0.5 to 2.5 K/s and TL peak maximum T_m was determined from the TL glow curves. k is Boltzmann constant. The linear fitting result is included in each subplot.

Fig. 4f. Calculated trap depth as a function of Sr content in the series of $\text{Ca}(\text{Sr})\text{ZnOS:Cu/Y}$ samples.

5. It is not clear enough how to obtain the results of Figure 5h. What is the meaning of patterned filter? Did the author use the synthesized PersL phosphors as lighting sources in the two right panels?

Response: We apologize for any confusion. The multicolor display using the as-prepared PersL phosphors is achieved by charging a flat-panel PersL film (PersL crystals encapsulated in PDMS matrix) covered by a patterned PET film. As shown in **Fig. 5a**, the PET film with a

predefined graphic acts as a photomask, selectively attenuating the charging light based on the target display shape. After charging, without removing the PET film, a multicolor graphic becomes visible. Thus, the PET film serves as both a mask at the charging stage and a color filter at the display stage.

For the proof-of-concept demonstration, we initially achieved super broadband persistent luminescence (PersL) emission in mixed Ca(Sr)ZnOS:Cu⁺/Y³⁺ crystals (with Ca_{0.75}Sr_{0.25}ZnOS:Cu⁺/Y³⁺ and Ca_{0.45}Sr_{0.55}ZnOS:Cu⁺/Y³⁺ at a weight ratio of 2:1). These crystals were then used to fabricate the PersL film for display applications. In **Fig. 5h**, the rightmost panel is based on Ca(Sr)ZnOS:Cu⁺/Y³⁺ crystals, while the middle panel serves as a comparison and is based on Ca(Sr)S:Eu²⁺—a red PersL phosphor with a narrow emission band.

6. I would like to suggest the authors to move Supplementary Figure 18 into the main figures because this is important to understand the charge transfer process in the studied materials.

Response: We are grateful for your valuable suggestion. We have incorporated the schematic illustration of PersL mechanism into **Fig. 4g** of the revised manuscript.

Accordingly, we have added the following discussion in the revised manuscript:

"The continuously decreasing trap depth suggests a more significant reduction in the band edge with respect to trap/donor levels. As a result, a global trap model is utilized, conceptualizing trap depth as the relative energy of V_s levels to the conduction band (**Fig. 4g**)."

Fig. 4g. Band diagram illustration of the PersL mechanism.

Reviewer #3 (Remarks to the Author):

1. The authors report on the luminescence properties of a particular phosphor based on (Ca,Sr)ZnOS:Cu,Y, where tuning of the luminescence (and of the persistent luminescence) can be obtained by changing the Ca-Sr composition. Also a study on the partial substitution of S by Se is mentioned, but this is less relevant, as an increase in the Se concentration quickly reduces the trapping capacity of the phosphor.

Response: We greatly appreciate your insightful feedback, and we wish to emphasize the core achievement of our research. As detailed in the manuscript, we have developed a method for continuously tuning persistent luminescence (PersL) within a single material system, for the first time. To more effectively showcase our material's superiority compared to those documented in existing literature, we have included a summary table (**Supplementary Table 1**) that encapsulates earlier attempts at PersL tuning. The investigation revealed that the conventional approach to modulating PersL typically involves combining various emission centers—such as triplet states, guest chromophores, and inorganic activators—within one host matrix. While these methods do result in distinct emission wavelengths, **they inevitably leave spectral gaps in specific wavelength ranges**, restricting the full spectrum emission of PersL. As outlined in the manuscript, we discovered that the PersL emission peak can be finely adjusted in a near-linear fashion across an extensive wavelength span (500–635 nm), with a remarkable precision of approximately 5 nm. This discovery permits the on-demand tailoring of PersL color (spectrum) spanning a wide spectral range of over 100 nm, **which is unparalleled in any other material system**.

The findings related to $\text{Ca}_{0.45}\text{Sr}_{0.55}\text{ZnOS}(\text{Se})\text{:Cu/Y}$ are presented with the intent of demonstrating the predominance of sulfur on PersL. It has been observed that PersL diminishes progressively as the sulfur content decreases (more S is substituted by Se). Although this aspect is not the main focus, it serves as an additional insight to comprehend the origin of PersL within $\text{Ca}(\text{Sr})\text{ZnOS:Cu/Y}$ systems. We hope you concur.

Supplementary Table 1. Brief summary of literature work on PersL spectrum tuning.

Materials	Emissions	Spectrum tunability	Ref.
Polycyclic aromatic hydrocarbons (PAHs)-melamine formaldehyde (MF) composites	466 nm (TP-MF) 503 nm (PA-MF) 544 nm (FA-MF) 592 nm (Py-MF) 601 nm (BA-MF)	Discrete	Angew. Chem. Int. Ed. 2024, 136, e202318516
Fluorescein sodium (FluNa)- $\text{Al}_2(\text{SO}_4)_3$	483, 549 nm	Discrete	Angew. Chem. Int. Ed. 2023, 135, e202217616
Calcein sodium (CalNa)- $\text{Al}_2(\text{SO}_4)_3$	466, 548 nm	Discrete	Angew. Chem. Int. Ed. 2023, 135, e202217616
MCATMA/C545T	408, 526, 560, 580 and 600 nm	Discrete	Adv. Mater. 2022, 34, 2206712
Cytosine-Cd/ZnX ₂ (X = Cl, Br)	435, 513 nm (Cy-CdCl ₂) 435, 513 nm (Cy-ZnCl ₂) 414 nm (Host)	Discrete	Adv. Sci. 2022, 9, 2200992
Poly(acrylamide-co-N-vinylcarbazole) based host-guest composites	502 nm (PBD, guest) 560 nm (Fluc, guest) 570 nm (Rh123, guest) 620 nm (RhB, guest)	Discrete	Sci. Adv. 2022, 8, eabk2925
(R,R)-DAACH/(S, S)-DAACH	470–530 nm	Discrete (Excitation dependent)	Nat. Commun. 2022, 13, 429

4,4'-bis(N-carbazolyl)-1,1'-biphenyl (pCBP)	430, 560 nm	Discrete	Angew. Chem. Int. Ed. 2020, 59, 10032
2,4,6-trimethoxy-1,3,5-triazine (TMOT)	465–505 nm	Discrete (Excitation dependent)	Nat. Photonics 2019, 13, 406
Organic molecules (H-aggregates)	530, 575 nm (DPhCzT) 515, 547 nm (DEOPh) 529, 574 nm (DECzT) 543, 591 nm (CzDCIT) 587, 644 nm (DCzPhP)	Discrete	Nat. Mater. 2015, 14, 685
Multicomponent copolymer	445, 517 nm (PDNA) 445, 514 nm (PDBA)	Discrete	Nat. Commun. 2020, 11, 944
NaLuF ₄ :Ln ³⁺ /Gd ³⁺ (Ln ³⁺ : trivalent lanthanide ion)	489, 546 and 584 nm (Tb ³⁺) 385 nm (Nd ³⁺) 453 nm (Tm ³⁺) 542 nm (Ho ³⁺) 543 nm (Er ³⁺) 573 nm (Dy ³⁺) 594 nm (Sm ³⁺) 606 nm (Pr ³⁺)	Discrete	Nature 2021, 590, 410
Pr ³⁺ doped in multiple hosts	UVC (Sr ₂ Al ₂ SiO ₇) 266 nm (Sr ₃ Y ₂ Si ₆ O ₁₈) 268 nm (Ca ₂ Al ₂ SiO ₇) 267 nm (Ca ₃ Al ₂ Si ₃ O ₁₂) 270 nm (Lu ₂ SiO ₅) 267 nm (LiYSiO ₄)	Discrete	Nat. Commun. 2020, 11, 2040
LiGa ₅ O ₈ :Mn ²⁺ transparent glass ceramic	510, 625 nm	Discrete	Light Sci. Appl. 2020, 9, 22
CdSiO ₃ @SiO ₂ nanoparticles	438 nm (In ³⁺) 438, 580 nm (In ³⁺ and Mn ²⁺) 549 nm (Tb ³⁺) 578 nm (Dy ³⁺) 620 nm	Discrete	Adv. Mater. 2020, 32, 2003881
SrSi ₂ O ₂ N ₂ :Ln ²⁺ /Ln ³⁺ (Ln: lanthanides)	(Yb ²⁺ /Dy ³⁺ , Yb ²⁺ /Ho ³⁺ , Yb ²⁺ /Er ³⁺) 540 nm (Eu ²⁺ /Dy ³⁺ , Eu ²⁺ /Ho ³⁺ , Eu ²⁺ /Er ³⁺)	Discrete	ACS Appl. Mater. Interfaces 2018, 10, 1854
NaYF ₄ :Ln ³⁺ @ NaYF ₄ (Ln: lanthanides)	1525 nm (Er) 1475 nm (Tm) 1180 nm (Ho) 1064 nm (Nd)	Discrete	Nat. Nanotechnol. 2021, 16, 1011
Cs ₂ CdCl ₄ and Cs ₂ CdCl ₄ :Mn ²⁺	500, 612 nm (Cs ₂ CdCl ₄) 600 nm (Cs ₂ CdCl ₄ :Mn ²⁺)	Discrete	Angew. Chem. Int. Ed. 2023, 62, e202308420
CaZnOS:Pb ²⁺ /Ln ³⁺ and CaZnOS:Cu ⁺ /Ln ³⁺	Multiple emission centers	Discrete	Laser Photonics Rev. 2023, 17, 2300132
CaAl ₂ O ₄ :Eu ²⁺ Nd ³⁺ /CsPbX ₃ /PDMS composite (X: Cl, Br, I)	440–694 nm (quantum dots as color converters)	Continuous (complex composite)	Angew. Chem. Int. Ed. 2019, 131, 7017

2. One of the reasons for this study is mentioned in the introduction: PersL emitters of various emission colors usually exhibit unbalanced brightness and efficiency. Unfortunately, this is also the case in the present solution: changing the Ca-Sr ratio also changes the trapping behaviour (as witnessed in the TL glow curves and in the intensity of the decay curves (Fig. 3). In that sense, the present phosphor system is not a real solution, where spectral tuning can be achieved independently of the trapping behaviour.

Response: Thank you for raising this point. Following an accurate control of the TL measurement conditions (charging wavelength/time, delay time, heating rate etc.), we find that altering the Ca/Sr ratio has a minor impact on the trapping behaviour of the PersL phosphor (**Fig. 4e**). The TL spectral profile remains largely unchanged, although its peak maximum slightly shifts to a lower temperature with the addition of Sr. On one side, As previously addressed, the primary focus of this work is the continuous tuning of PersL wavelength in a single material system. On another, "brightness" is a concept determined by human visual perception, which is inherently influenced by wavelength. Even in the well-established CsPbX₃ (X=Cl, Br, I) system, achieving emission with equal brightness across a spectrum of colors remains a challenge. To more accurately reflect the storage capability of our PersL phosphors, we have employed an integrated TL intensity as a figure of merit. The result is depicted in **Fig. R2**, where the coefficient of variation of integrated TL intensity across eight samples is 24.95%, endorsing the achievement of "wavelength-tunable PersL with comparable emission intensity". To the best of our knowledge, no prior research has shown the ability to finely tune PersL across a range of wavelengths while maintaining uniform trapping behaviours and comparable storage capacities. We hope you concur.

Fig. R2. Integrated TL intensity as function of Sr content.

3. Of course, the study by itself contains interesting results, but I am not convinced that the proposed material system will have a high impact in the field, given that the aforementioned problem is not really solved. In the application part, a 'super broadband PersL' is created by mixing two Ca-Sr compositions. Doesn't this create the same issue, regarding color changes during the decay?

Response: We appreciate your recognition of the strengths of our study. As outlined in **Supplementary Table 1**, color tuning on PersL is a constant subject of research. Therefore, we believe that achieving continuous tuning of PersL with high wavelength accuracy in a single material system will be of significant interest to the field.

Furthermore, we are confident that color variation will not be an issue for the proposed demonstration. For confirmation, we recorded the PersL spectrum of the two-component mixture to check whether it will change over an extended period, with the findings detailed in **Fig. R3**. The broadband PersL spectrum exhibits negligible shifts over time, effectively mitigating any worries regarding color alteration throughout its operation. We hope you concur.

Fig. R3. The stability of the PersL spectra for the two-component mixture.

4. Furthermore, in the abstract it is claimed that the high initial brightness is up to 5 cd/m^2 , although I could not find data supporting this claim. All the data for the decay curves are shown in arbitrary units, except for supplementary Figure 9, but there the initial intensity is lower than 5 cd/m^2 . Furthermore, it is mentioned that it was measured with a (scanning) spectrometer, calibrated with a luminance meter. More details should definitely be provided on the calibration method, as the luminance (measured in cd/m^2) depends on the shape of the spectrum. Hence, a calibration of the FLS980 spectrometer to yield cd/m^2 values for widely different emission colors is not straightforward. The advise would be to measure the phosphors directly, after excitation, with the luminance meter, and then compare them. Preferably, also standard excitation conditions (Xe lamp and D65 lamp) should be used to give a fair estimate of the obtainable brightness.

Response: We regret any misunderstanding caused. The initial brightness refers to the luminance value (captured by a CHROMA METER CS-200 luminance meter) **5 seconds** after turning off the charging light (a 4W 365 nm lamp for 3 minutes). The luminance values measured at different delay moments for the Ca(Sr)/ZnOS:Cu/Y samples are detailed in **Supplementary Table 3**. However, owing to the luminance meter's detection limit being 0.01 cd/m^2 —significantly higher than the threshold value 0.032 mcd/m^2 —it was not possible to accurately determine the PersL duration of our materials. To rectify this, it is essential to calibrate the PersL decay curves using the recorded luminance values. The calibration assumed that the luminance follows the same trend as the PersL decay curves, and the charging light source and delay time are kept the same for both PersL decay and luminance measurements.

The decay curves of PersL were recorded using the FLS980 spectrometer, set to kinetic scan mode, which tracks the intensity decay at a specific wavelength. In a detailed measurement process, the sample was initially charged with a 365 nm handheld lamp (4W) for 3 minutes. Subsequently, the sample was placed inside the spectrometer's chamber. A fixed delay of 20 seconds was allowed between the end of the charging and the commencement of the measurement. For luminance calibration, the luminance value at 20 s after the ceasing of optical charging were utilized. This corresponds to a value of 1.14 cd/m^2 for

Ca_{0.6}Sr_{0.4}ZnOS:0.1%Cu⁺/1%Y³⁺ as indicated in **Supplementary Fig. 10** and documented in **Supplementary Table 3** in the revised Supplementary Information.

In accordance with your recommendation, we employed a D65 lamp as a charging light source, which has been proven to be able to excite CaZnOS:Cu/Y crystals. However, due to the limited spectral overlap between the emission of the lamp and the PersL excitation of our PersL material (**Supplementary Fig. 8**), the luminance values were lower than those previously measured. The results are compiled into the **Supplementary Table 4**.

Furthermore, our materials have demonstrated the capability to emit **higher brightness** when subjected to a high-power excitation source. In our experiments, we charged the Ca(Sr)ZnOS:Cu/Y crystals using a 10 W 365 nm lamp. The resulting brightness values shown below in **Table R1**. Notably, a peak luminance of 12.11 cd/m² was observed at 5 seconds after the excitation source was turned off.

Table R1. The brightness of the newly developed Ca(Sr)ZnOS PersL materials (Charged by a 10 W 365 nm lamp).

	Brightness (cd/m ²)				
	5 s	10 s	20 s	30 s	1 min
CaZnOS:Cu/Y	6.72	2.43	1.02	0.61	0.28
Ca _{0.9} Sr _{0.1} ZnOS:Cu/Y	8.40	3.09	1.28	0.78	0.36
Ca _{0.75} Sr _{0.25} ZnOS:Cu/Y	10.22	3.75	1.59	0.99	0.45
Ca _{0.6} Sr _{0.4} ZnOS:Cu/Y	12.11	4.19	1.69	1.06	0.48
Ca _{0.45} Sr _{0.55} ZnOS:Cu/Y	10.99	3.91	1.59	0.95	0.45
Ca _{0.3} Sr _{0.7} ZnOS:Cu/Y	9.38	3.20	1.30	0.78	0.36
Ca _{0.15} Sr _{0.85} ZnOS:Cu/Y	4.93	1.75	0.72	0.44	0.21
SrZnOS:Cu/Y	3.73	1.30	0.49	0.31	0.14

We have added the following statement in the revised manuscript:

"Particularly, these materials are distinguished by their remarkable ability to be excited by a standard D65 lamp (**Supplementary Fig. 8 and Supplementary Table 4**), considerably broadening their applicability in lighting, displays and safety indications."

We have added additional information regarding the PersL decay curves and luminance calibration in the revised Supplementary Information:

"The decay curves of PersL were recorded using the FLS980 spectrometer, set to kinetic scan mode, which tracks the intensity decay at a specific wavelength. In a detailed

measurement process, the sample was initially charged with a 365 nm handheld lamp (4W) for 3 minutes. Subsequently, the sample was placed inside the spectrometer's chamber. A fixed delay of 20 seconds was allowed between the cessation of the charging and the commencement of the measurement. For luminance calibration, the luminance value at 20 s after ceasing the optical charging was utilized. The calibration assumed that the luminance follows the same trend as the PersL decay curves, and the charging light source and delay time are kept the same for both PersL decay and luminance measurements."

Supplementary Fig. 8. PersL excitation spectrum of CaZnOS:Cu/Y compared to the emission spectrum of a commercial D65 lamp.

Supplementary Table 4. The brightness of the newly developed Ca(Sr)ZnOS PersL materials (Charged by an 18 W D65 lamp).

	Brightness (cd/m ²)				
	5 s	10 s	20 s	30 s	1 min
CaZnOS:Cu/Y	0.48	0.28	0.17	0.13	0.09
Ca _{0.9} Sr _{0.1} ZnOS:Cu/Y	0.66	0.40	0.23	0.18	0.10
Ca _{0.75} Sr _{0.25} ZnOS:Cu/Y	1.27	0.74	0.41	0.30	0.17
Ca _{0.6} Sr _{0.4} ZnOS:Cu/Y	1.97	1.04	0.55	0.39	0.21
Ca _{0.45} Sr _{0.55} ZnOS:Cu/Y	2.11	1.12	0.62	0.45	0.25
Ca _{0.3} Sr _{0.7} ZnOS:Cu/Y	1.59	0.85	0.46	0.34	0.19
Ca _{0.15} Sr _{0.85} ZnOS:Cu/Y	0.86	0.44	0.24	0.17	0.11
SrZnOS:Cu/Y	1.39	0.63	0.31	0.21	0.11

5. It would also be interesting to plot in Fig. 3d the peak and fwhm not only for the PL, but also for the PersL, and discuss in more detail the differences.

Response: Thanks for your kind suggestion. We have updated the peak and FWHM for both PL and PersL (**Fig. 3d**). The peak wavelength of PL is shorter (corresponding to higher emission energy) than that of PersL for the series of CaZnOS:Cu/Y crystals and the FWHM values of PL are larger than the correspondent PersL. To understand the difference, we consider the transition energy of D-A pair luminescence, which can be expressed as:

$$E = E_g - (E_D + E_A) + \frac{e^2}{4\pi\epsilon r}$$

where E_g is the bandgap energy, E_D/E_A is the ionization energy of the donor/acceptor, ϵ is the dielectric constant of the crystal and r is the distance between D-A pair. During photoexcitation, the average distance between donor-acceptor pairs significantly decreases. This reduction in distance results in an increase in recombination energy E , which consequently leads to a decrease in the emission wavelength when compared to that of PersL. In a similar situation, photoexcitation is responsible for a wider distribution of donor-acceptor pair distances, which manifests as a broader bandshape relative to PersL.

We have added the following discussion in the revised manuscript:

"However, the peak wavelength of PL is observed to be slightly shorter compared to that of PersL across the series of Ca(Sr)ZnOS:Cu/Y crystals. This phenomenon is attributed to the significantly reduced average distance between D-A pairs upon photoexcitation, leading to higher emission energy.

Along with the redshifted emission peak, the emission band of PL and PersL are obviously broadened, with the full width at half-maximum (FWHM) varying from 117 to 182 nm for PL and 117 to 170 for PersL (**Fig. 3d**). The difference in the FWHM between PL and PersL originated from the broader distribution of donor-acceptor (D-A) pair distances in the presence of photoexcitation, which results in a more substantial FWHM for PL when compared to PersL."

Fig. 3. Linear tuning of PersL in Ca(Sr)ZnOS:Cu⁺/Y³⁺. **a**, PL spectra of Ca_{1-x}Sr_xZnOS:0.1%Cu⁺/1%Y³⁺ ($x = 0-1$) crystals. **b**, PersL spectra of Ca_{1-x}Sr_xZnOS:0.1%Cu⁺/1%Y³⁺ ($x = 0-1$) after turning off the UV light (4 W, 365 nm). **c**, PersL photographs of Ca(Sr)ZnOS:0.1%Cu⁺/1%Y³⁺ with various Sr contents. **d**, PL/PersL peak wavelength (top) and FWHM (bottom) as a function of Sr concentration in the Ca(Sr)ZnOS:0.1%Cu⁺/1%Y³⁺ crystals. **e**, PersL decay profiles of Ca(Sr)ZnOS:0.1%Cu⁺/1%Y³⁺ with various Sr contents. The samples were pre-charged using a 365 nm UV lamp (4 W) for 3 min and a short delay of 20 s was allowed before each measurement.

6. There is an issue with the TL curves in Figure 4. The erratic way in which the plot lines are shown does not have a meaning (a TL measurement assumes a constant heating rate, now the curves jump back and forth as a function of temperature). Also, it is mentioned that the TL glow curves are nearly identical, but the differences are actually quite large and they do not change in a very systematic way when the composition is changed. For a publication in Nat Comm, this should be investigated in more detail.

Response: We regret any misunderstanding that may have arisen. We have re-tested the TL glow curves with stringent control over the experimental conditions, including pre-charging wavelength/time, delay time, and heating rate. Furthermore, we have updated our TL measurement apparatus to the LTTL-3DS, a domestic TL/OSL spectrometer. This instrument has the ability to discern wavelength details, enriching our comprehension of the TL signal by introducing an extra dimension of analysis. When we integrate the data across wavelengths, we obtain the traditional two-dimensional TL glow curve, illustrating the relationship between TL intensity and temperature.

The results indicate that the general shape of the TL glow curves remains consistent, yet there is a minor shift in the peak maxima towards a lower temperature (from 359 K to 348 K at a heating rate of 1 K/s) correlating with increased Sr concentrations. This supports our hypothesis that Sr alloying does not modify the trap distribution but does lead to a systematic reduction in trap depth values. These insights have been integrated into **Fig. 4e** in the revised manuscript. Additionally, we have included the original three-dimensional contour plots of the TL spectra in **Supplementary Fig. 21**. Owing to their consistent emission characteristics, TL also exhibits a spectral profile that can be finely adjusted, similar to that of room temperature PersL.

We have added the following discussion in the revised manuscript:

"However, the maximum of the TL glow peak shifts to a lower temperature (from 359 K to 348 K), indicating a consistent decrease in the trap depth. Additionally, the observed shift in the TL emission peak, in correlation to an increase in Sr content, is consistent with the behaviour noted in room-temperature PersL, indicating a similar nature between TL and PersL (**Supplementary Fig. 21**)."

Fig. 4. Mechanistic investigations of D-A luminescence characteristics in Ca(Sr)ZnOS:Cu/Y crystals. **a**, The calculated location of defect states related to the Fermi level in $\text{Ca}_{1-x}\text{Sr}_x\text{ZnOS}:\text{Cu}^+/\text{Y}^{3+}$ ($x = 0, 0.4, 0.7$ and 1). **b**, Comparison of the energy difference (between the selected defect levels) and D-A emission energy in Ca(Sr)ZnOS. **c-d**, PersL spectra and PersL decay curves of the $\text{Ca}_{0.45}\text{Sr}_{0.55}\text{ZnOS}_{1-x}\text{Se}_x:\text{Cu}/\text{Y}$ ($x = 0, 0.25, 0.5, 0.75$ and 1) crystals. **e**, TL spectra of $\text{Ca}_{1-x}\text{Sr}_x\text{ZnOS}:0.1\%\text{Cu}^+/1\%\text{Y}^{3+}$ ($x = 0, 0.1, 0.25, 0.4, 0.55, 0.7, 0.85$ and 1) after charging at 365 nm (4 W). The heating rate is 1 K/s. **f**, Calculated trap depth as a function of Sr content in the series of Ca(Sr)ZnOS:Cu/Y samples. **g**, Band diagram illustration of the PersL mechanism.

Supplementary Fig. 21. Contour mapping of TL intensity as a function of emission wavelength and temperature for $\text{Ca}_{1-x}\text{Sr}_x\text{ZnOS:0.1\%Cu}^+/1\%\text{Y}^{3+}$ ($x = 0, 0.1, 0.25, 0.4, 0.55, 0.7, 0.85$ and 1). The heating rate during TL measurement is 1 K/s . The shift in the emission peak with increasing Sr content aligns with the behaviour observed in room temperature PersL, indicating a similar luminescence nature between TL and PersL.

7. I will not comment on the theoretical calculations, as I am not an expert in the field. Regarding the conclusion from the Se substitution, it is a bit surprising to read that the PersL is strongly correlated with sulfur, while it is actually a vacancy. This should be explained in more detail.

Response: We apologize for any confusion. The observation that systematically substituting sulfur (S) with selenium (Se) leads to a consistent reduction in PersL, while preserving the PL of D-A emission, highlights sulfur's critical influence on the PersL characteristics. This suggests that the trapping process is intimately associated with sulfur. Considering the minimal role of interstitial sulfur in trapping/emission (please refer to Fig. 4b), it is logical to deduce that sulfur vacancies primarily govern the PersL. Our further calculations indicate the

unlikelihood of selenium vacancies functioning as trapping centers, given their energy levels lie outside the energy bandgap, which explains the compromised PersL performance after Se doping. On a side note, recent studies have shown that sulfur vacancies can significantly affect the photoluminescence of materials like molybdenum disulfide (MoS₂). These vacancies create defect states within the band gap, which can trap excitons and lead to emissions at specific energies, observable even at room temperature (*ACS Nano* **2023**, *17*, 13545). Such findings provide strong support for our conclusions. We hope you concur.

We have corrected the corresponding statement in the revised manuscript:

"This implies a close association between the trapping state and sulfur. Given the negligible involvement of interstitial S in the trapping/emission mechanisms (**Fig. 4b**), it can be inferred that sulfur vacancies are the principal determinants of PersL. Our theoretical calculations indicate that selenium vacancies are unlikely to serve as trapping centers, as their energy levels are positioned outside the inter-bandgap, affirming the critical role of V_S in the PersL process (**Supplementary Fig. 20**)."

8. Finally, I must admit that – even after reading several times - I didn't get how the programmable optical storage would actually be used in practice. In panel f, other information appears compared to panel e (Figure 5), but how does this change evolve over time, and what is then the "useful" time period to do the read out? I do not see where an optical information storage during 5s (or 60s) would be useful for. Also for the multicolor display, there is some uncertainty about what is exactly done. Was the phosphor excited through the patterned film? Or would such a 'display' be excited without the pattern, after which the user inserts the film? Furthermore, is the emission spectra stable through the mentioned duration of 6 hours? Is it then still possible to see color information?

Response: We apologize for any confusion. To enhance the practicality of the proposed optical storage, we employ thermoluminescence signals to extend the duration for information retrieval. The new design encompasses photomask-based charging and TL readout. Here, the thermoluminescence intensity varies markedly with different photomasks (70% versus 40%) over an extended period. However, after a delay of 12 hours, these differences become negligible (**Supplementary Fig. 28**). By capitalizing on this characteristic, we use the luminescence intensity contrast for information storage and successfully prolonged the operational lifespan of optical storage to 8 hours (**Fig. 5d-f**). The evolution of the information is shown in **Supplementary Fig. 29**. This advancement holds potential for implementation in the fresh food supply chain to inspect product freshness. We hope you concur.

Fig. 5. Ca(Sr)ZnOS:Cu/Y PersL crystals for information applications. **a**, Schematic illustration of photomask charging technique using flat-panel PersL thin film. **b**, Comparison of PersL decay of the $\text{Ca}_{0.45}\text{Sr}_{0.55}\text{ZnOS}:0.1\%\text{Cu}^+/1\%\text{Y}^{3+}$ sample covered by photomasks with various grayscale values during charging (100% and 0% stand for total transmission and depletion of charging light, respectively). **c**, TL spectra of the $\text{Ca}_{0.45}\text{Sr}_{0.55}\text{ZnOS}:0.1\%\text{Cu}^+/1\%\text{Y}^{3+}$ in the presence of various grayscale photomasks during charging. The samples were pre-charged using a 365 nm UV lamp (4 W) for 5 s and a short delay of 20 s was allowed before each measurement. **d-f**, Demonstration of programmable information coding based on charging through a patterned photomask. The grayscale values of the photomask are 10% and 40/70% for charging blocked and partially blocked parts, respectively. The luminescence intensity contrast is used for information encoding. The partially blocked regions (40/70%) remain distinguishable after an 8-hour delay after charging. **g**, Super broadband PersL spectrum of blended $\text{Ca}(\text{Sr})\text{ZnOS}:\text{Cu}^+/\text{Y}^{3+}$ crystals ($\text{Ca}_{0.75}\text{Sr}_{0.25}\text{ZnOS}:\text{Cu}^+/\text{Y}^{3+}$ and $\text{Ca}_{0.45}\text{Sr}_{0.55}\text{ZnOS}:\text{Cu}^+/\text{Y}^{3+}$ at a weight ratio of 2:1). The spectrum of red-emitting $\text{Ca}(\text{Sr})\text{S}:\text{Eu}^{2+}$ with narrow band is presented for comparison. **h**, Multicolor display through PersL film containing blended $\text{Ca}(\text{Sr})\text{ZnOS}:\text{Cu}^+/\text{Y}^{3+}$ crystals in the presence of a patterned filter, which shows overwhelming color resolvability compared to the PersL film based on conventional $\text{Ca}(\text{Sr})\text{S}:\text{Eu}^{2+}$.

Supplementary Fig. 28. TL spectra of $\text{Ca}_{0.45}\text{Sr}_{0.55}\text{ZnOS}:0.1\% \text{Cu}^+/1\% \text{Y}^{3+}$ (Charged for 5 s; 70% or 40% grayscale photomask) with various delay times (20 s, 2 h, 4 h, 8 h and 12 h) before each measurement.

Supplementary Fig. 29. Time evolution of optical information through thermoluminescence readout. The luminescence intensity contrast of partially blocked regions (40/70%) remain distinguishable after an 8-hour delay following charging.

Regarding concerns on the multicolor display, the phosphor was excited through the patterned film. The multicolor display using the as-prepared PersL phosphors is achieved by charging a flat-panel PersL film (PersL crystals encapsulated in PDMS matrix) covered by a patterned PET film. As shown in **Fig. 5a**, the PET film with a predefined graphic acts as a photomask, selectively attenuating the charging light based on the target display shape. After charging, without removing the PET film, a multicolor graphic becomes visible. Thus, the PET film serves as both a mask at the charging stage and a color filter at the display stage.

The duration of 6 hours is inferred from the PersL duration test as outlined in the response to Q4 (charged with a 4W 365 nm UV source). At a luminance below 0.032 mcd/m^2 , it's unlikely to see any color information.

Since our PersL materials are demonstrated to be excitable by the standard D65 light source, we proposed a charging scheme that harnesses both PersL and the D65 illuminant. In this setup, the PersL film is covered by a patterned film on one side, while the D65 light source is positioned on the opposite side. Upon D65 illumination, the resulting image reveals a distinct pattern, as presented in the **Supplementary Fig. 30**. When the D65 light is switched off, the pattern remains visible to the naked eye. This design holds great promise for large-scale billboard applications during nighttime, where a noncontinuous illumination scheme can be adopted, offering an effective solution for energy conservation.

Supplementary Fig. 30. D65 charged multicolor display based on the PersL film containing blended $\text{Ca}(\text{Sr})\text{ZnOS}:\text{Cu}^+/\text{Y}^{3+}$ crystals. a) Schematic of the illuminating setup, the PersL film is covered by the patterned film on one side, while the D65 light source is positioned on the opposite side for charging. b) Photograph of the display when D65 lamp is turned on. c) Photograph of the display after D65 lamp is turned off.

Accordingly, we have added the following discussion in the revised manuscript:

"Benefiting from the capability of excitation by the D65 lamp, the multicolor display holds great promise for large-scale billboard applications during nighttime, where a noncontinuous illumination scheme can be adopted, offering an effective solution for energy conservation (Supplementary Fig. 30)."

Reviewer #1 (Remarks to the Author):

Since the authors have revised the manuscript appropriately in response to my comments, I would recommend its publication for Nature Communications.

Reviewer #2 (Remarks to the Author):

The revised version of the manuscript looks good, and I have no additional comments on it. Therefore, I recommend it to be published in the current form.

Reviewer #3 (Remarks to the Author):

The authors have answered in great detail to the reviewer comments. The issues were clarified and new measurements were performed and discussed. One of the main issues, regarding the afterglow intensity, was clearly resolved.